# The genetic architecture of protein stability

Andre J. Faure[1,5 ✉], Aina Martí-Aranda[1,2], Cristina Hidalgo-Carcedo[1], Antoni Beltran[1], Jörn M. Schmiedel[1,6] & Ben Lehner[1,2,3,4 ✉]

There are more ways to synthesize a 100-amino acid (aa) protein ($20^{100}$) than there are atoms in the universe. Only a very small fraction of such a vast sequence space can ever be experimentally or computationally surveyed. Deep neural networks are increasingly being used to navigate high-dimensional sequence spaces[1]. However, these models are extremely complicated. Here, by experimentally sampling from sequence spaces larger than $10^{10}$, we show that the genetic architecture of at least some proteins is remarkably simple, allowing accurate genetic prediction in high-dimensional sequence spaces with fully interpretable energy models. These models capture the nonlinear relationships between free energies and phenotypes but otherwise consist of additive free energy changes with a small contribution from pairwise energetic couplings. These energetic couplings are sparse and associated with structural contacts and backbone proximity. Our results indicate that protein genetics is actually both rather simple and intelligible.

Massively parallel experiments allow the effects of single aa changes in proteins to be comprehensively quantified[2,3]. Similarly, experimental analysis of double mutants is feasible, at least for small proteins[4,5]. The analysis of higher-order mutants, however, quickly becomes infeasible owing to the combinatorial explosion of possible genotypes. For example, the number of ways to combine one mutation at 34 different sites in a protein is $2^{34} \approx 1.7 \times 10^{10}$. Experimental exploration of such a large number of genotypes is extremely challenging[6] given current technology, which—so far—has experimentally analysed sequence spaces up to about $10^6$ (refs. 4,7).

Moreover, combining random mutations in even moderate numbers nearly always results in non-functional proteins[8,9]. For example, only 2–8% of 5 aa variants and <0.2% of 10 aa variants in a small protein domain are expected to be folded if energies combine additively ($n = 2$ domains; Fig. 1a and Extended Data Fig. 1a). Sampling even tens of millions of random combinatorial genotypes in most proteins will therefore provide almost no information about genetic architecture—the set of rules that govern how mutations combine to determine phenotypes—and will not be useful for training and evaluating predictive models beyond testing the trivial prediction that most genotypes are unfolded.

One strategy for exploring high-dimensional sequence spaces is to use deep learning. Deep neural networks with millions of fitted parameters have proved successful for diverse prediction and protein design tasks, including predicting the effects of combinatorial mutants[10–21]. However, these models have extremely complicated and difficult to interpret architectures.

It could be that protein genotype–phenotype landscapes are complex, with many interactions between mutations required for accurate prediction. Alternatively, these landscapes might be much simpler, as suggested by energy measurements[22] and inferences[23,24] and the use of statistical models[25,26]. For us, a simple model is one with few parameters (so providing a large data compression) and parameters that are interpretable (so providing understanding).

Here we use an experimental design that enriches functional protein sequences to explore the genetic architecture of high-dimensional protein sequence spaces with more than 30 dimensions and more than $10^{10}$ genotypes. We find that protein architectures are remarkably simple, with additive energy models providing very good predictive performance. Quantifying the pairwise energetic couplings between mutations further increases predictive power, providing excellent performance in high-dimensional genotype spaces. These couplings are sparse and related to protein 3D structures. The genetic architecture of at least some proteins is therefore very simple, with additive energetics and a small contribution from sparse pairwise structural couplings.

## Sampling a $10^{10}$ sequence space

We previously showed that the energetic effects of thousands of individual mutations on the stability of a protein can be measured en masse using pooled variant synthesis, selection and sequencing experiments[23,24]. In these experiments, the effect of each mutation on the cellular abundance of a protein is measured in the wild-type protein and also in a small number of variants with different fold stabilities. For example, using a shallow double-mutant library, we could infer the changes in Gibbs free energy of folding ($\Delta\Delta G_f$) for nearly all mutations (1,056/1,064 = 99%) in the C-terminal SH3 domain of the adaptor protein GRB2 (ref. 23). Similar massively parallel measurements of single-mutant fold stabilities have now been made for other signalling domains, including the oncoprotein KRAS[24], and, in vitro, for many, mostly prokaryotic, small domains[7].

Combining random mutations in the GRB2-SH3 domain very quickly results in unfolded proteins, with about 98% and more than 99.9% of genotypes with five and ten mutations expected to be unfolded (based on additive energies; see Fig. 1a). This rapid decay of stability as mutations are combined is consistent with experimental measurements of the activity of other proteins[8,9]. To experimentally explore folded

[1]Centre for Genomic Regulation (CRG), The Barcelona Institute of Science and Technology, Barcelona, Spain. [2]Wellcome Sanger Institute, Wellcome Genome Campus, Hinxton, UK. [3]Universitat Pompeu Fabra (UPF), Barcelona, Spain. [4]Institució Catalana de Recerca i Estudis Avançats (ICREA), Barcelona, Spain. [5]Present address: ALLOX, Barcelona, Spain. [6]Present address: factorize.bio, Berlin, Germany. ✉e-mail: andre.faure@crg.eu; bl11@sanger.ac.uk

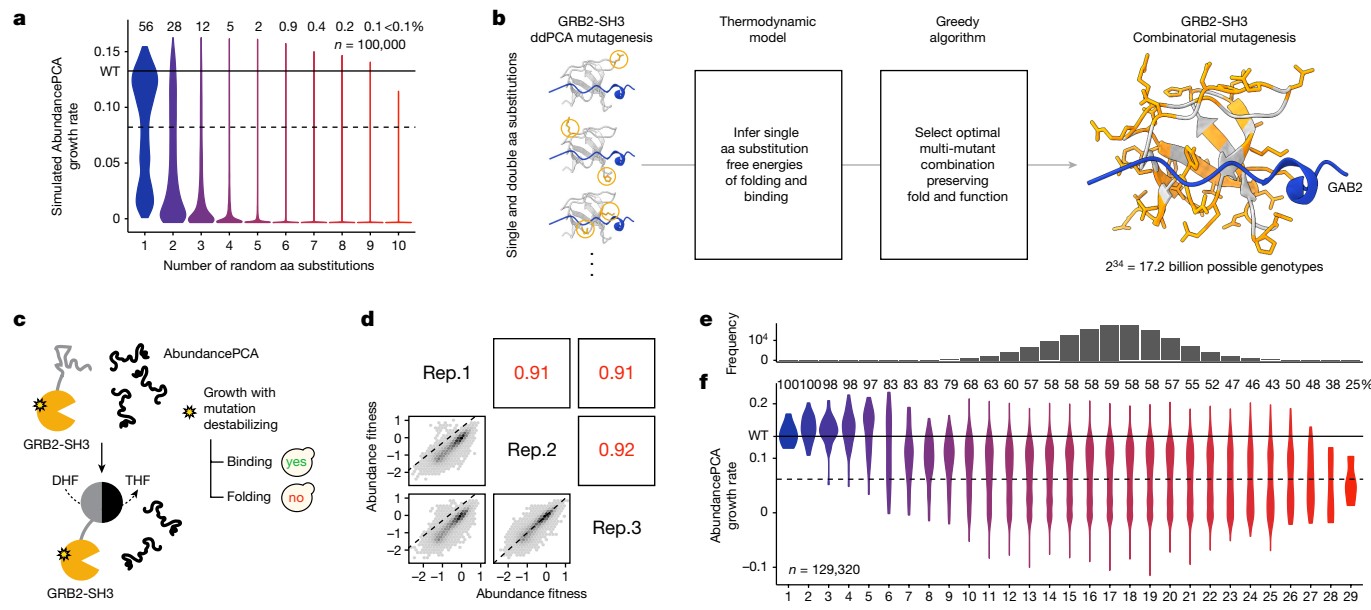

**Fig. 1 | An efficient strategy to explore high-dimensional protein sequence space and enrich multi-mutants for conserved fold and function. a**, Violin plot showing distributions of simulated AbundancePCA growth rates (assuming additivity of individual inferred folding free energy changes[23]) versus number of random aa substitutions ($n = 100,000$). Violins are scaled to have the same maximum width. **b**, DMS data, energy model and algorithm used to select a set of single aa substitutions for combinatorial mutagenesis. A shallow double-mutant library of GRB2-SH3 protein variants was assayed by AbundancePCA (see panel **c**) and BindingPCA (see Fig. 4b; in combination referred to as ddPCA), followed by energy modelling to infer single aa substitution free energy changes of folding and binding[23]. We used this model together with a greedy algorithm to select a set of 34 single aa substitutions that, when combined, would simultaneously maximize both the predicted AbundancePCA and BindingPCA growth rates, that is, preserving both fold

and function. 3D structure of GRB2-SH3 (PDB: 2VWF) indicating the 34 combinatorially mutated residues (orange) and GAB2 ligand (blue) is shown on the right. **c**, Overview of AbundancePCA on the protein of interest (GRB2-SH3)[23]. yes, yeast growth; no, yeast growth defect; DHF, dihydrofolate; THF, tetrahydrofolate. **d**, Scatter plots showing the reproducibility of fitness estimates from triplicate AbundancePCA experiments. Pearson's $r$ is indicated in red. Rep., biological replicate. **e**, Histogram showing the number of observed aa variants at increasing Hamming distances from the wild type (denoted by WT), for which the $x$ axis is shared with panel **f**. **f**, Violin plot showing distributions of AbundancePCA growth rates inferred from deep sequencing data versus number of aa substitutions. In panels **a** and **f**, the percentage of folded protein variants (predicted fraction folded molecules > 0.5) is shown at each Hamming distance from the wild type.

genotypes in high-dimensional sequence spaces, we therefore used a heuristic technique to enrich for conserved fold and function in combinatorial variants. For each possible starting single aa substitution, we iteratively selected further substitutions—one per residue position—that simultaneously maximizes the resulting combinatorial mutant's predicted abundance and binding to an interaction partner (see Methods). For GRB2-SH3, the largest set of mutations predicted to preserve both molecular phenotypes consisted of 34 single aa substitutions: 25 in surface residues (relative solvent-accessible surface area (RSASA) ≥ 0.25), three in the protein core (RSASA < 0.25) and six mutations in the GAB2 ligand binding interface (ligand distance < 5 Å; Fig. 1b, right).

We synthesized a library ('library 1') containing all combinations of these 34 mutants and quantified the cellular abundance of a sample of the $2^{34} \approx 1.7 \times 10^{10}$ genotypes using a highly validated pooled selection, abundance protein fragment complementation assay (AbundancePCA[23,24,27]). In total, we obtained triplicate abundance measurements for 129,320 variants, which is 0.0007% of the sequence space. The measurements were highly reproducible (Pearson's $r > 0.91$; Fig. 1d and see Methods).

The symmetrical pod-like shape of the genotype frequency landscape, with the number of genotypes peaking at the intermediate Hamming distance of 17—that is, equidistant from the wild type (zeroth-order) and 34th-order mutant—is recapitulated in the experimentally sampled library (Fig. 1e). Median abundance measurements decrease with increasing number of aa substitutions, but there are still thousands of genotypes with many mutations that, nevertheless, maintain abundance scores that are indistinguishable from that of the

wild-type protein ($n = 2,706$ with more than 20 mutations, two-sided $z$-test nominal $P > 0.05$; Fig. 1f).

## Genetic prediction with energy models

Quantifying the effects of a large number of multi-mutants allowed us to test the predictive performance of genotype–phenotype models in regions of the genetic landscape beyond the local neighbourhood used for training. For model building and evaluation, we restricted all analyses to variants with quantitative measurements in all three biological replicates ($n = 71,233$). Notably, our original energy model (Fig. 2a) trained on abundance and ligand binding selections (doubledeepPCA, ddPCA) quantifying the effects of single and double aa mutants only[23] explains as much as half of the fitness variance in combinatorial multi-mutants ($R^2 = 0.5$; Fig. 2b, lower-right panel), for which most (94%) include at least 13 aa substitutions in the wild-type sequence. The only trained parameters in this simple model are Gibbs free energy terms for the wild type ($\Delta G_f$) and single aa substitutions ($\Delta\Delta G_f$) and a two-parameter (affine) transformation relating the fraction of folded molecules to the AbundancePCA score (fitness; Fig. 2a). That such a large proportion of phenotypic variance is explained by an additive energy model (no specific epistasis/genetic interactions) trained on genotypes containing only one or two genetic changes suggests that the energetic effects of mutations in proteins are largely context-independent.

On the other hand, a linear model—which implicitly assumes that mutation effects combine additively at the phenotypic level in multi-mutants—trained on the same ddPCA data performs much worse

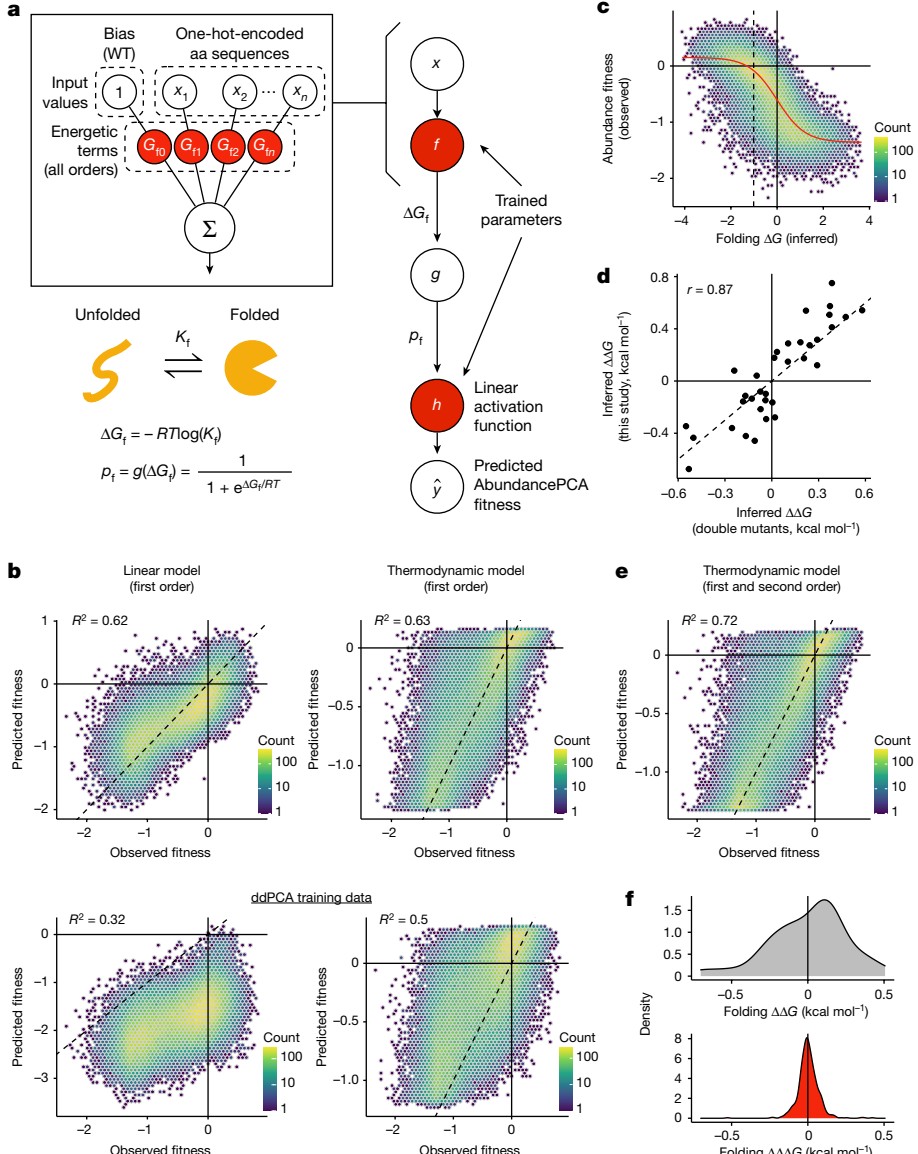

**Fig. 2 | Thermodynamic modelling of protein abundance to infer folding free energy changes and energetic couplings. a**, Two-state equilibrium and corresponding neural network architecture used to fit thermodynamic model to AbundancePCA data (bottom, target and output layer), thereby inferring the causal changes in free energy of folding associated with single aa substitutions (top, input values). $\Delta G_f$, Gibbs free energy of folding; $K_f$, folding equilibrium constant; $p_f$, fraction folded; $g$, nonlinear function of $\Delta G_f$; $R$, gas constant; $T$, temperature in kelvin. **b**, Performance of first-order linear models (left column) and first-order energy models (right column) evaluated on GRB2-SH3 combinatorial AbundancePCA data. The top row indicates the results of models that were trained on a subset of the same combinatorial DMS data. The bottom row indicates the results of models that were trained on GRB2-SH3

ddPCA data consisting of single and double aa substitutions only[23]. $R^2$ is the proportion of variance explained. **c**, Nonlinear relationship (global epistasis) between observed AbundancePCA fitness and changes in free energy of folding. Thermodynamic model fit is shown in red. **d**, Comparisons of the model-inferred free energy changes to previously reported estimates using GRB2-SH3 ddPCA data[23]. Pearson's $r$ is shown. **e**, Performance of energy model that includes all first-order and second-order genetic interaction (energetic coupling) terms/coefficients. See Extended Data Fig. 2 for plots of the residuals versus fitted values for linear and energy models of the first and second order. **f**, Distributions of folding free energy changes ($\Delta\Delta G$, grey) and pairwise energetic couplings ($\Delta\Delta\Delta G$, red). WT, wild type.

($R^2 = 0.32$). The linear model also systematically underestimates the observed phenotypic effects of mutant combinations (Fig. 2b, lower-left panel), a consequence of not accounting for the scaling of mutational effects owing to protein thermodynamics (global epistasis[28–30]). For example, introducing a destabilizing mutation in an already-unfolded protein has no effect on the fraction of folded molecules (lower plateau of model in Fig. 2c), which is not captured by a linear model. These results demonstrate a key advantage of fitting energy models: accounting for global epistasis improves the generalizability of predictions beyond the local neighbourhood of the training data.

Fitting linear and energy models to the combinatorial data improves the variance explained by 30% and 13%, respectively (Fig. 2b, upper panels), probably because of the greater amount of training data and (relevant) genetic backgrounds in which the effects of each single aa are quantified: about 50% of all variants in the library ($n \approx 30,000$) include—and therefore report on the effects of—any given one of the 34 single aa substitutions, that is, almost three orders of magnitude more measurements per single mutant compared with the relatively shallow (ddPCA) library. Although the fraction of variance explained by first-order linear and energy models is comparable ($R^2 = 0.62$ and 0.63;

Fig. 2b, upper panels), the biased regression residuals in the case of the linear model show that this model is less appropriate (Extended Data Fig. 2a). The energy model provides an excellent fit to the data, faithfully capturing the global nonlinear relationship (global epistasis) between observed AbundancePCA fitness and inferred changes in free energy of folding ($\Delta G_f$) (Fig. 2c). There is also very good agreement between free energy changes (model parameters) inferred from combinatorial and ddPCA datasets (Pearson's $r = 0.87$), but the former tend to be more extreme, once again demonstrating the use of assaying the effects of mutations in greater numbers of genetic backgrounds, thereby allowing their energetic effects to be more accurately estimated (Fig. 2d).

## Couplings improve genetic prediction

We next tested whether quantifying non-additive energetic couplings between mutations improved predictive performance. In our combinatorial dataset, each pair of mutations is present in a median of 17,923 genotypes, allowing robust measurement of second-order genetic interaction terms (energetic couplings, $\Delta\Delta\Delta G_f$ (refs. 31,32)). Including all second-order energetic couplings improves model performance by an extra 9% ($R^2 = 0.72$), consistent with expectations that pairwise effects are an important source of specific epistasis in proteins[32] (Fig. 2e). Whereas first-order terms are stronger in magnitude and biased towards destabilizing effects, second-order energetic couplings tend to have milder effects centred on zero (Fig. 2f).

## Physical contacts and backbone proximity

Measured at the phenotypic level, genetic interactions in proteins have previously been shown to reflect—at least in part—protein structures[4,33–36]. Combining combinatorial deep mutational scanning with thermodynamic modelling allowed us to infer a total of 561 pairwise energetic couplings, providing an opportunity to interrogate their mechanistic origins and relationship with protein structure. Comparing coupling energy magnitude (absolute folding $\Delta\Delta\Delta G_f$) with the 3D distance separating mutation pairs in the folded structure (minimal side-chain heavy-atom distance) reveals an L-shaped distribution with the strongest energetic couplings occurring between structurally proximal residues (Fig. 3a; see also Extended Data Fig. 3a). The top five energetic couplings all involve pairs of residues within 5.5 Å and 15 of the top 20 (75%) energetic couplings involve pairs separated by less than 8 Å. Although there is a weak anticorrelation between contact distance and coupling energy strength (Spearman's $\rho = -0.12$; Fig. 3a), this trend breaks down for pairs that are not proximal in the primary sequence (Spearman's $\rho = -0.02$, backbone distance >5 residues).

On the other hand, comparing coupling strength with separation distance between residues in the primary sequence (along the peptide backbone) reveals a marked inverse relationship that extends over quite large distances (Spearman's $\rho = -0.28$) and is robust to the exclusion of direct physical contacts between residues (<5 Å, Spearman's $\rho = -0.27$; Fig. 3b and see also Extended Data Fig. 3b). The interaction matrix in Fig. 3c summarizes these observations: the strongest energetic couplings coincide with direct physical contacts (black circles; see also Fig. 3d) and energetic coupling strength decays along the protein backbone (Fig. 3c, near-diagonal versus far off-diagonal cells). The matrix also highlights physical interactions between secondary structural elements as hotspots for strong energetic couplings.

To disentangle the relative importance of these different potential structural determinants of energetic coupling strength, we gathered a collection of quantitative features describing both the number and the type of chemical bonds or interactions existing between the atoms of pairs of residues, as well as their relative positions in the folded structure (Fig. 3e). A linear regression model based on these 12 structural features is predictive of coupling strength (Fig. 3f; see Methods). Notably, the same model performs similarly well when evaluated on a held-

out, non-overlapping set of inferred energetic couplings derived from an independent combinatorial mutagenesis experiment ('library 3', which is described below; Pearson's $r = 0.46$, $R^2 = 0.21$). This suggests that, despite its simplicity, the integrated model captures structural determinants of energetic coupling strength and that energetic couplings are caused by structural interactions.

## Couplings decay along the peptide chain

To directly test the hypothesis that inter-residue backbone distance is associated with energetic coupling strength independently of 3D contact distance, we designed a combinatorial saturation mutagenesis library involving all possible mutations at four physically proximal surface residues in the same secondary structure element ('library 2'; Extended Data Figs. 4a and 5a and see Methods). We reasoned that energetic couplings owing to the propagation of perturbations along the protein backbone should also be apparent among solvent-facing residues. In total, we obtained abundance measurements for 138,157 variants (86% of the sequence landscape) and the measurements were highly reproducible (Pearson's $r > 0.89$; Extended Data Fig. 5b and see Methods).

The single-mutant effects at these four residues have a larger range than those of the combinatorial library that was designed to conserve fold and function (Extended Data Fig. 5c,d). Therefore, when combined in double, triple and quadruple mutants—the most numerous class—the result is a larger fraction of unfolded variants (Extended Data Fig. 5c–e). A two-state thermodynamic model that includes all first-order and second-order coefficients provides an excellent fit to the data ($R^2 = 0.93$; Extended Data Fig. 5e–g) and inferred folding free energy changes (first-order terms) are highly correlated (Pearson's $r = 0.94$) with those obtained previously using an independent shallow double-mutant library (Extended Data Fig. 5h). Although the four mutated residues are physically proximal in 3D space, with all except one pair (H26:T44) separated by less than 5 Å (3.8–8.4 Å; Extended Data Fig. 5i), their relative positions in the primary peptide sequence cover a large range (2–18 residues; Extended Data Fig. 5j). There is no relationship between contact distance and folding coupling strength for these contacting residues (Spearman's $\rho = -0.05$; Extended Data Fig. 5i), whereas the relationship for backbone distance is significant (Spearman's $\rho = -0.41$; Extended Data Fig. 5j). Indeed, backbone distance is very well correlated with coupling strength when averaging energy terms per residue pair (Spearman's $\rho = -0.94$; Extended Data Fig. 5j). The relative position of aa residues in the primary protein sequence is therefore associated with coupling strength independently of their proximity in 3D space.

## Higher-order mutants fold and function

Our experiments identified a large number of GRB2-SH3 genotypes containing many mutations that have high cellular abundance (for example, 25,564 genotypes containing more than five mutations; Fig. 1f). To further confirm that abundant multi-mutants are correctly folded and functional, we performed a third combinatorial mutagenesis experiment in which we also tested the ability of GRB2-SH3 variants to bind to a peptide ligand using a protein–protein interaction assay (BindingPCA[23,24,37]; Fig. 4a). Recognition of the peptide ligand can only occur if the protein adopts its native conformation[38] (Fig. 1b,c). We designed a library (library 3) consisting of all combinations of 15 single aa substitutions occurring within a 22-aa residue window, avoiding mutations in our original library in binding interface residues (minimal side-chain heavy-atom distance to the ligand < 5 Å; Extended Data Fig. 4b and also see Methods). The library contains $2^{15}$ (=32,768) variants and shares six single aa substitutions with our original $2^{34}$ library. In total, we obtained binding measurements for 25,967 variants and abundance measurements for 31,936 variants (79% and 97% of the sequence

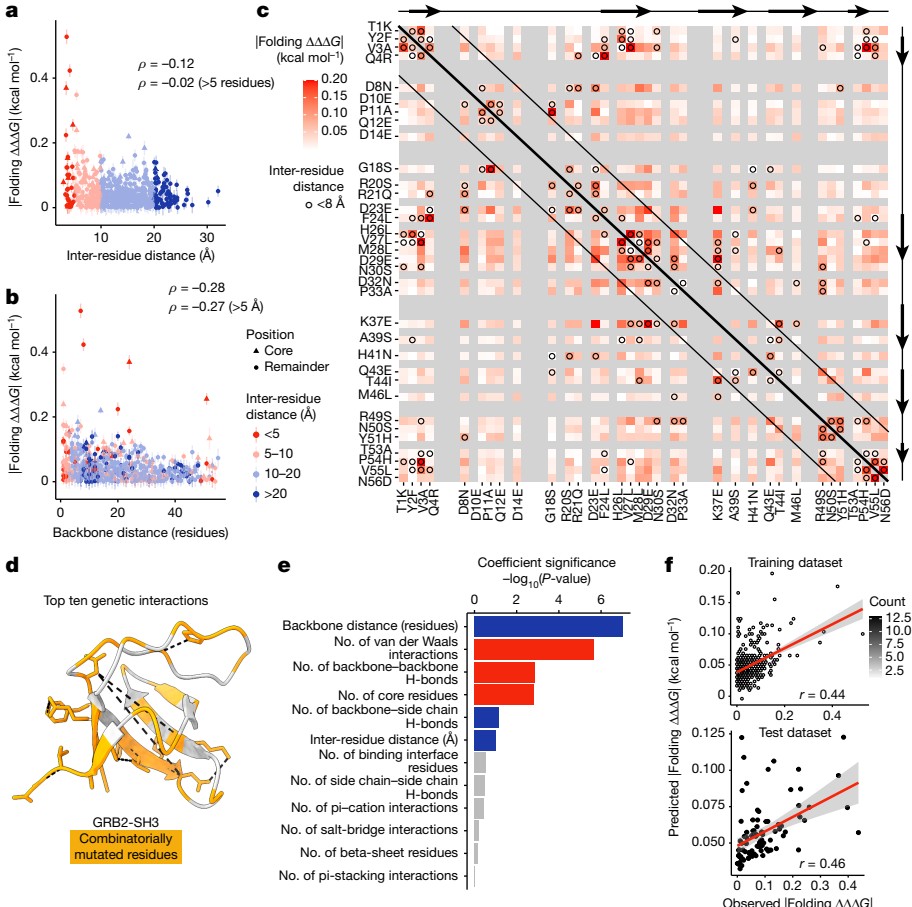

**Fig. 3 | Structural determinants of energetic couplings inferred from combinatorial mutagenesis. a**, Relationship between folding coupling energy strength and minimal inter-residue side-chain heavy-atom distance. The mean is shown and error bars indicate 95% confidence intervals from a Monte Carlo simulation approach ($n = 10$ experiments). Points are coloured by binned inter-residue distances (see legend in panel **b**). Spearman's $\rho$ is shown for all couplings (top value), as well as those involving pairs of residues separated by more than five residues in the primary sequence (bottom value). Core residues are indicated as triangles. **b**, Relationship between folding coupling energy strength and linear sequence (backbone) distance in number of residues. The measure of centre and error bars are as defined in panel **a**. **c**, Interaction matrix indicating folding energy coupling strength, as well as pairwise structural contacts in GRB2-SH3 (Protein Data Bank (PDB): 2VWF; minimal side-chain heavy-atom distance < 8 Å, black circles). Grey cells indicate

missing values (non-mutated residues) and constitutive single aa substitutions are indicated in the *x*-axis and *y*-axis labels. Black diagonal lines demarcate residue pairs that are distal in the primary sequence (backbone distance > 5 residues). Reference secondary structure elements (arrow, beta strand) are shown at the top and right. **d**, 3D structure of GRB2-SH3 (PDB: 2VWF) indicating the top ten energetic couplings with black dashed lines. Combinatorially mutated residues are shown in orange. **e**, Bar plot showing ranked features from the linear model to predict folding coupling strength. Bar width indicates coefficient significance (*P* value from uncorrected two-sided *t*-test). Blue, positive coefficient; red, negative coefficient; grey, non-significant (*P* > 0.05). **f**, Correlation between linear model predicted and observed folding coupling strength for training (top) and test (bottom) data. Pearson's *r* is shown. The error bands represent the 95% confidence intervals for the predicted values.

landscape, respectively). The measurements were highly reproducible (Pearson's *r* > 0.85 and 0.94 for binding and abundance, respectively; Fig. 4c, Extended Data Fig. 6a–f and also see Methods).

Plotting the changes in binding against the changes in abundance for third-order, sixth-order and ninth-order variants shows that most mutations altering binding also alter the concentration of the isolated domain, consistent with previous results and the expectation that changes in protein stability are a main cause of mutational effects on binding[23,24,39] (Fig. 4d and Extended Data Fig. 6g). Notably, however, most higher-order mutants that have high abundance scores also bind the GAB2 ligand, indicating that they are correctly folded (Fig. 4d and Extended Data Fig. 6g). For example, 4% (204/4,805) of variants containing nine mutations have abundance indistinguishable from that of the wild-type protein (nominal *P* > 0.05) and 96% (177/184) of these also bind the ligand (predicted fraction bound molecules > 0.5). Most of the abundant higher-order GRB2-SH3 mutants are thus correctly folded.

## Multi-phenotype genetic prediction

The large number of genetic backgrounds in which both single and double aa mutant effects were measured for these two related molecular phenotypes is a rich source of data for thermodynamic modelling. First, considering only the abundance phenotype, we observe that an additive two-state thermodynamic model—with unfolded and folded energetic states—outperforms a linear model when evaluated on held-out variants ($R^2 = 0.93$ versus 0.87; Extended Data Fig. 7a,b). To attain similar predictive performance as the first-order energy model requires inclusion of both second-order and third-order genetic interaction terms in the linear model (Extended Data Fig. 7c,d), representing a massive increase in model complexity (715 versus 16 parameters; that is, greater than 40-fold more). This greater complexity of models that use many specific pairwise and higher-order genetic interaction terms to capture global nonlinearities in data (global epistasis) has been referred to as 'phantom epistasis'[40].

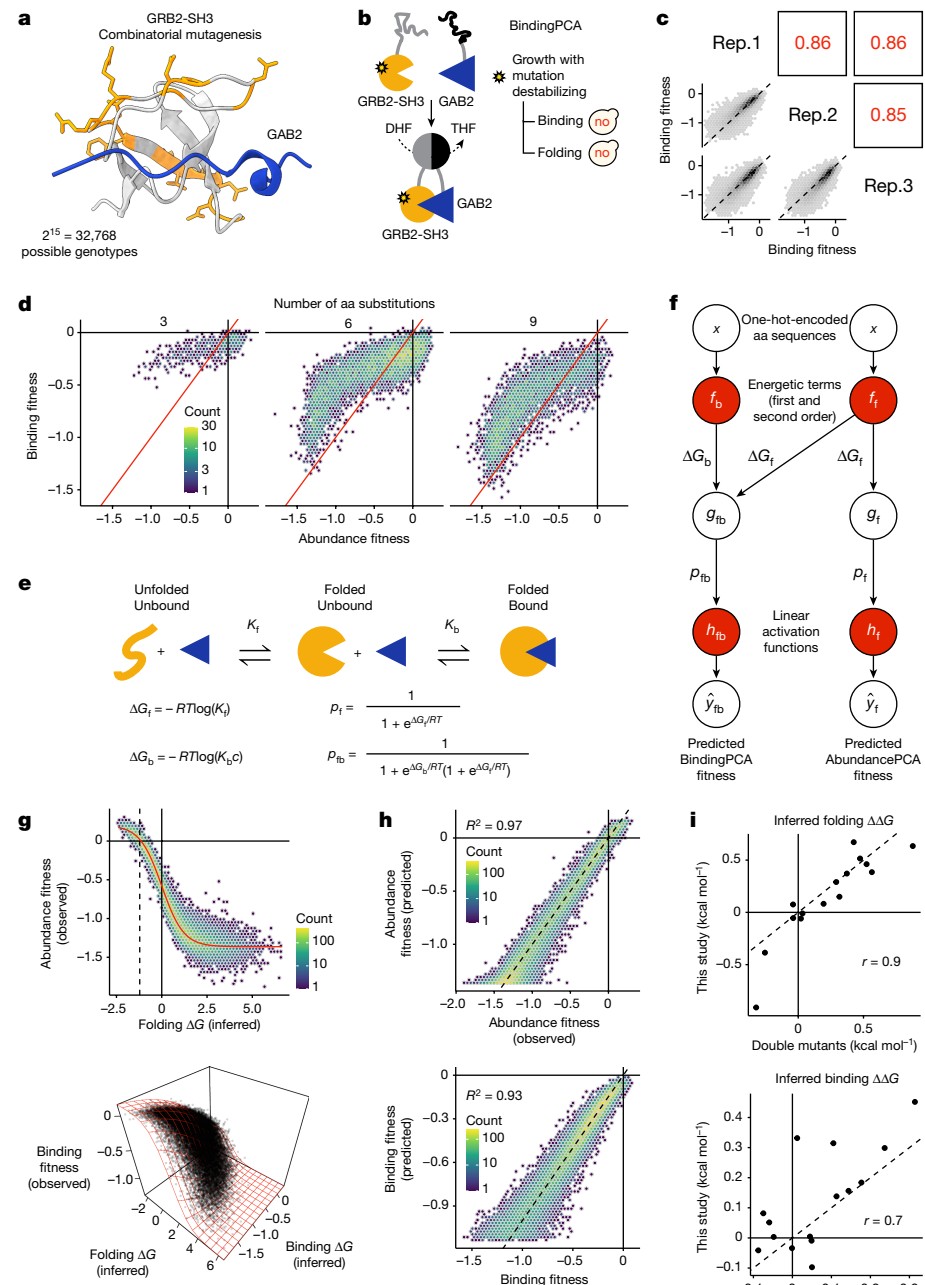

**Fig. 4 | Combinatorial ddPCA shows that abundant multi-mutants are binding-competent (have a conserved fold). a**, 3D structure of GRB2-SH3 (PDB: 2VWF) indicating 15 combinatorially mutated residues in library 3 (orange) and GAB2 ligand (blue). **b**, Overview of BindingPCA of GRB2-SH3 binding to GAB2 (ref. 23). no, yeast growth defect; DHF, dihydrofolate; THF, tetrahydrofolate. **c**, Scatter plots showing the reproducibility of fitness estimates from triplicate BindingPCA experiments. Pearson's *r* indicated in red. Rep., biological replicate. **d**, 2D density plots comparing abundance and binding fitness of third-order (left), sixth-order (middle) and ninth-order mutants (right). See Extended Data Fig. 6g for similar plots at all mutant orders. **e**, Three-state equilibrium and corresponding thermodynamic model. $\Delta G_f$, Gibbs free energy of folding; $\Delta G_b$, Gibbs free energy of binding; $K_f$, folding equilibrium constant; $K_b$, binding equilibrium constant; $c$, ligand concentration; $p_f$, fraction folded; $p_{fb}$, fraction folded and bound; $R$, gas

constant; $T$, temperature in kelvin. **f**, Neural network architecture used to fit thermodynamic model to ddPCA data (bottom, target and output data), thereby inferring the causal changes in free energy of folding and binding associated with single aa substitutions (first-order terms) and pairwise (second-order) interaction terms (top, input values). Variables as per panel **e** and: $g_f$, nonlinear function of $\Delta G_f$; $g_{fb}$, nonlinear function of $\Delta G_f$ and $\Delta G_b$. **g**, Nonlinear relationships between observed AbundancePCA fitness and changes in free energy of folding (top) or BindingPCA fitness and free energies of both binding and folding (bottom). Thermodynamic model fit is shown in red. **h**, Performance of models fit to ddPCA data. $R^2$, proportion of explained variance. **i**, Comparisons of the model-inferred single aa substitution free energy changes to previously reported estimates using GRB2-SH3 ddPCA data[23]. Pearson's *r* is shown.

Next, extending previous work[23], we used a neural network implementation of a three-state equilibrium model[41]—with unfolded, folded and bound energetic states (Fig. 4e)—to simultaneously infer the underlying causal free energy changes of both folding and binding ($\Delta\Delta G_f$

and $\Delta\Delta G_b$), as well as folding and binding energetic couplings ($\Delta\Delta\Delta G_f$ and $\Delta\Delta\Delta G_b$) (Fig. 4f). The model fits the data extremely well (Fig. 4g), explaining virtually all of the fitness variance (Fig. 4h), and the inferred folding and binding free energy changes (first-order terms) are well

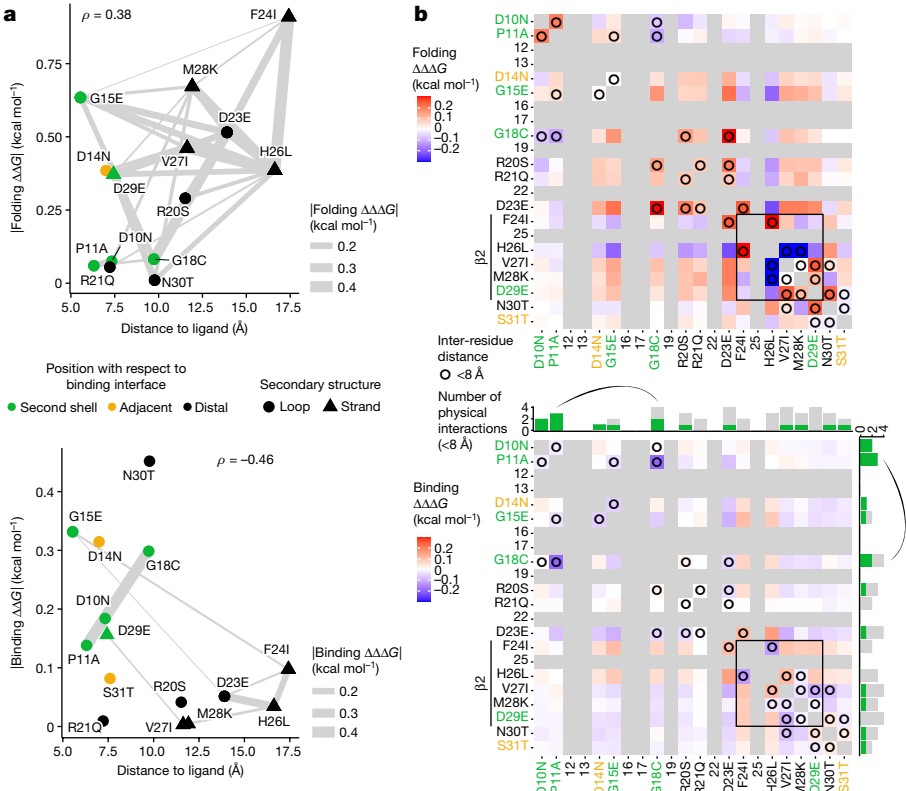

**Fig. 5 | Proximity and connectivity of residues near the binding interface explain energetic effects on binding affinity. a**, Relationship between the absolute change in free energy of folding (top) and binding (bottom) and minimal side-chain heavy-atom distance to the ligand. Residues are coloured by their position in the structure relative to the binding interface, triangles indicate beta strand residues and connection lines indicate the strength of energetic couplings between aa pairs (see legend). Spearman's $\rho$ is shown. **b**, Interaction matrix indicating folding (top) and binding (bottom) coupling terms, as well as pairwise structural contacts in GRB2-SH3 (PDB: 2VWF; minimal

side-chain heavy-atom distance < 8 Å, black circles). Grey cells indicate missing values (non-mutated residues) and constitutive single aa substitutions are indicated in the *x*-axis and *y*-axis labels (see panel **a** for axis label text colour key). Mutations in beta strand residues are indicated and couplings between beta strand residues are boxed. The bar plots above and to the right of the binding interaction matrix indicate the total number of pairwise physical interactions (<8 Å) involving each residue, with green bars indicating the fraction of interacting partners classified as second-shell residues. The strongest binding energetic coupling (P11A:G18C) is indicated by an arc.

correlated (Pearsons's *r* = 0.9 and 0.7) with those obtained previously using an independent shallow double-mutant library[23] (Fig. 4i, double mutants). This is the first time, to our knowledge, that a large number of folding ($\Delta\Delta\Delta G_f$) and binding ($\Delta\Delta\Delta G_b$) energetic coupling terms have been measured for any protein.

## Allostery in ligand-proximal residues

We observe that mutational effects on folding energy tend to be larger than those on binding energy (Fig. 5a), recapitulating previous results[23,24]. Energetic couplings show the same pattern, with folding coupling energies tending to be larger in magnitude than binding energetic couplings (area under the curve = 0.7, *n* = 210, *P* = 3.6 × 10$^{-7}$, two-sided Mann–Whitney *U* test, |$\Delta\Delta\Delta G_f$| mean = 0.087, s.d. = 0.084, |$\Delta\Delta\Delta G_b$| mean = 0.038, s.d. = 0.035; Fig. 5b). As none of the mutations in this library occur in the binding interface, any notable effects on binding affinity must be through an allosteric mechanism[23,24]. Plotting absolute free energy changes against the 3D distance to the ligand shows a negative correlation as previously reported[23,24] (Spearman's $\rho$ = −0.46), with mutations in second-shell residues and residues adjacent (in the sequence) to binding interface residues highly enriched for strong allosteric effects on binding affinity (Fig. 5a). Consistent with previous observations[23], mutations at distal glycine residues have among the strongest effects on binding affinity.

Whereas the mutations with the strongest folding coupling energies are near-diagonal (closely spaced in the primary sequence), particularly

between pairs of residues in the beta strand, the strongest binding coupling in the dataset is an interaction between residues P11 and G18 (Fig. 5b). These two residues are proximal in 3D space (<8 Å) and constitute one of only two long-range physical contacts between the mutated residues (backbone distance > 5 residues; Fig. 5b), suggesting that allosteric energetic couplings are also driven by structural contacts.

## SRC kinase combinatorial mutagenesis

Finally, to further test the generality of our conclusions, we used the same greedy approach to design a library containing 2$^{15}$ (=32,768) variants in an unrelated and larger protein, the human proto-oncogene tyrosine-protein kinase Src (SRC). We obtained triplicate abundance measurements for 31,557 variants and the measurements were highly reproducible (*r* > 0.86; Fig. 6b–d). As for our three GRB2-SH3 combinatorial libraries, a second-order energy model was highly predictive of abundance changes ($R^2$ = 0.87; Fig. 6e,f), with energetic couplings predicted by both 3D spatial proximity (Fig. 6g) and backbone proximity (Fig. 6h). The consistency of these results in an unrelated full-length protein further supports their generality.

## Discussion

By experimentally quantifying protein fold stability in samples from sequence spaces greater than 10$^{10}$ in size, we have shown here that the fundamental genetic architecture of at least some proteins is

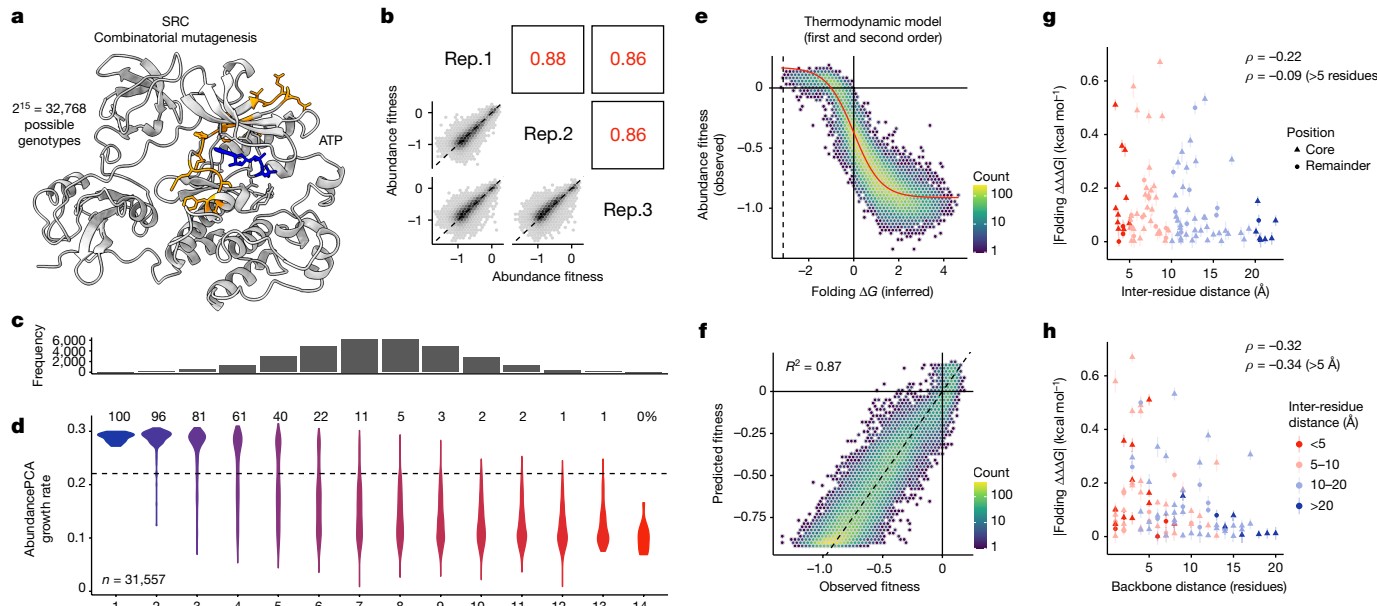

**Fig. 6 | Combinatorial mutagenesis of SRC. a**, 3D structure of SRC (PDB: 2SRC) indicating the 15 combinatorially mutated residues in library 4 (orange) and ATP (blue). **b**, Scatter plots showing the reproducibility of fitness estimates from triplicate AbundancePCA experiments. Pearson's *r* indicated in red. Rep., biological replicate. **c**, Histogram showing the number of observed aa variants at increasing Hamming distances from the wild type, in which the *x* axis is shared with panel **d**. **d**, Violin plot showing distributions of abundance growth rates inferred from deep sequencing data versus number of aa substitutions. The percentage of bound protein variants (predicted fraction bound molecules > 0.5) is shown at each Hamming distance from the wild type. **e**, Nonlinear relationship (global epistasis) between observed abundance fitness and changes in free energy of folding. Thermodynamic model fit is

shown in red. **f**, Performance of energy model that includes all first-order and second-order genetic interaction (energetic coupling) terms/coefficients. **g**, Relationship between folding coupling energy strength and minimal inter-residue side-chain heavy-atom distance. The mean is shown and error bars indicate 95% confidence intervals from a Monte Carlo simulation approach ($n = 10$ experiments). Points are coloured by binned inter-residue distances (see legend in panel **h**). Spearman's $\rho$ is shown for all couplings (top value), as well as those involving pairs of residues separated by more than five residues in the primary sequence (bottom value). Core residues are indicated as triangles. **h**, Relationship between folding coupling energy strength and linear sequence (backbone) distance in number of residues.

remarkably simple. Thermodynamic models in which the energetic effects of mutations are summed provide very good prediction of fold stability when tens of mutations are combined. Quantifying the pairwise energetic couplings between mutations further increases predictive power, providing very good performance in high-dimensional genotype spaces. The large number of energetic couplings quantified here reveals important principles about their origins: couplings are strongest between structurally contacting residues and coupling strength also decays along the protein backbone.

The energy models used here are very sparse and represent very large data compressions: up to about $10^8$ ($2^{34}/34$) for the additive models and up to about $10^7$ ($2^{34}/596$) for the models with energetic couplings. Analyses of previously published combinatorial protein mutagenesis datasets[8,41–43], mutagenesis of a protein interaction interface[44], hydrophobic protein cores[45], an intrinsically disordered region[46], a tRNA[43,47] and an alternatively spliced exon[40] suggest that this simplicity of genotype–phenotype landscapes is widely observed and probably a general principle of macromolecules and their molecular interactions.

Energy models are grounded in our understanding of protein thermodynamics and their simplicity and interpretability contrasts with the complexity and lack of mechanistic insight provided by deep neural networks. Predictive energy models are likely to have many applications, including for clinical variant effect interpretation[48], pathogen and pandemic forecasting[49] and protein engineering for biotechnology[1]. An important challenge moving forward is how to efficiently quantify the free energy changes and energetic couplings for all mutations in proteins of interest. Quantifying mutational effects across diverse genetic backgrounds and homologous sequences may be an efficient way to achieve this[50].

Our data do not rule out the importance of higher-order genetic interactions for protein stability. Rather, they show that, when global nonlinearities owing to cooperative protein folding are properly accounted for and measurements are averaged across genetic backgrounds, first-order and pairwise energetic couplings provide sufficient information for many prediction tasks. An important question to address in future work will be the extent to which higher-order energetic interactions become important in even larger sequence spaces, including in the 'twilight zone' of structurally homologous proteins with very low sequence identity. Indeed, the superior performance of our models in $2^{15}$-sized compared with in $2^{34}$-sized sequence spaces hints that higher-order interactions become increasingly important as sequences diverge. Simple experimental designs should be able to definitively address this question for a diversity of protein folds.

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

## Methods

### Combinatorial mutagenesis library designs

**Combinatorial library 1.** Library 1 was designed using a computationally efficient greedy strategy to search for the largest number of single aa substitutions that, when combined, preserve both fold and function even in the highest-order mutants (Fig. 1b). The algorithm used previously published ddPCA data and thermodynamic modelling results for GRB2-SH3, including inferred single aa substitution free energy changes of folding and binding for this protein[23]. We showed previously that this model—which assumes that individual inferred folding and binding free energy changes ($\Delta\Delta G_f$ and $\Delta\Delta G_b$) combine additively in multi-mutants—accurately predicts the effects of double aa substitutions[23]. Therefore, this same additive model was used to make predictions about the energetic and phenotypic effects of higher-order mutants explored in the greedy search.

First, the set of candidate single aa mutations was restricted to those with confident free energy changes, defined as those with 95% confidence intervals < 1 kcal mol$^{-1}$ and whose effects were measured in at least 20 genetic backgrounds (that is, double aa mutations). Candidate mutations were further restricted to those reachable by single-nucleotide substitutions in the wild-type sequence to simplify synthesis of the resulting combinatorial mutagenesis library. The algorithm begins from an arbitrary starting mutation and iteratively selects further mutations at other residue positions until all residues in the protein have been mutated. The heuristic works by selecting further mutations at each step that maximize the fold and function of the current highest-order mutant combination, that is, the geometric mean of model-predicted AbundancePCA and BindingPCA growth rates. This procedure is then repeated for all possible starting mutations.

To visualize and compare the resulting solutions, we also simulated the median AbundancePCA and BindingPCA growth rates of all candidate combinatorial libraries, calculated using a random sample of 10,000 variants. Although the algorithm is not guaranteed to produce the optimal solution at each Hamming distance from the wild-type sequence, the greedy approach nevertheless achieves solutions in which both phenotypes are predicted to be preserved in variants with more than 30 mutations (Extended Data Fig. 1b), beyond which one or both phenotypes are lost. Defining viable libraries as those preserving both molecular phenotypes above 70% of the maximal value (that is, the geometric mean of simulated median AbundancePCA and BindingPCA growth rates) resulted in the largest candidate combinatorial library consisting of all combinations of 34 single aa mutations (Fig. 1 and Extended Data Fig. 1b–d).

**Combinatorial library 2.** We clustered the contact map (minimal side-chain heavy-atom distance < 5 Å) comprising all GRB2-SH3 surface residues (RSASA ≥ 0.25) existing in secondary structure elements (Extended Data Fig. 4) and selected the following four physically proximal residues for saturation combinatorial mutagenesis: H26, M28, A39 and T44 (see Extended Data Fig. 5).

**Combinatorial library 3.** This library was designed to include all combinations of 15 single aa substitutions with mild effects (within one-third of the AbundancePCA fitness interquartile range of the wild type[23]) in close proximity in the primary sequence and reachable by single-nucleotide substitutions while avoiding mutations in binding interface residues (minimal side-chain heavy-atom distance to the ligand < 5 Å). We used a sliding window approach to determine the number of candidate mutant residues in stretches of 20, 21 and 22 consecutive residues in GRB2-SH3 (Extended Data Fig. 4b). Only one window with a width of 22 aa (starting at residue position 10) includes 15 candidate positions (Extended Data Fig. 4b). The final library consisted of all combinations of the following randomly selected candidate

mutations at these positions: D10N, P11A, D14N, G15E, G18C, R20S, R21Q, D23E, F24I, H26L, V27I, M28K, D29E, N30T and S31T (see Fig. 4).

**Combinatorial library 4: SRC.** This library was designed using the same greedy algorithm from data and thermodynamic modelling results for SRC[51], including inferred single aa substitution free energy changes of folding and activity for this protein. The design includes 15 single aa substitutions reachable by single nt substitution in a 22 aa window, located in the N-lobe of the SRC kinase domain, avoiding mutations in the activation loop, subsetting folding and activity ddGs to confident energies (95% confidence interval < 1 kcal mol$^{-1}$) and associated with singles observed in at least seven backgrounds. The final library consisted of all combinations of the following randomly selected candidate mutations at these positions: V329G, G344S, F349V, K343M, E331K, V337A, E332A, M341K, S330N, I336L, T338S, S345T, L346V, P333T and Y340S (see Fig. 6).

### Mutagenesis library construction and selection assays
#### Media and buffers used.

- LB: 10 g l$^{-1}$ bacto-tryptone, 5 g l$^{-1}$ yeast extract, 10 g l$^{-1}$ NaCl. Autoclaved 20 min at 120 °C.
- YPDA: 20 g l$^{-1}$ glucose, 20 g l$^{-1}$ peptone, 10 g l$^{-1}$ yeast extract, 40 mg l$^{-1}$ adenine sulphate. Autoclaved 20 min at 120 °C.
- SORB: 1 M sorbitol, 100 mM LiOAc, 10 mM Tris pH 8.0, 1 mM EDTA. Filter sterilized (0.2-mm nylon membrane, Thermo Scientific).
- Plate mixture: 40% PEG3350, 100 mM LiOAc, 10 mM Tris-HCl pH 8.0, 1 mM EDTA pH 8.0. Filter sterilized.
- Recovery medium: YPD (20 g l$^{-1}$ glucose, 20 g l$^{-1}$ peptone, 10 g l$^{-1}$ yeast extract) + 0.5 M sorbitol. Filter sterilized.
- SC-URA: 6.7 g l$^{-1}$ yeast nitrogen base without aa, 20 g l$^{-1}$ glucose, 0.77 g l$^{-1}$ complete supplement mixture drop-out without uracil. Filter sterilized.
- SC-URA/MET/ADE: 6.7 g l$^{-1}$ yeast nitrogen base without aa, 20 g l$^{-1}$ glucose, 0.74 g l$^{-1}$ complete supplement mixture drop-out without uracil, adenine and methionine. Filter sterilized.
- Competition medium: SC-URA/MET/ADE + 200 µg ml$^{-1}$ methotrexate (Merck Life Science), 2% DMSO.
- DNA extraction buffer: 2% Triton-X, 1% SDS, 100 mM NaCl, 10 mM Tris-HCl pH 8.0, 1 mM EDTA pH 8.0.

**Plasmid construction.** For libraries 1–3: GRB2 mutagenesis plasmid pGJJ286: wild-type GRB2-SH3 was digested from pGJJ046 (described previously[23]) with the restriction enzymes AvrII and HindIII and cloned into the digested plasmid pGJJ191 (described previously[24]) using T4 ligase (New England Biolabs). AbundancePCA pGJJ046 and pGJJ045 plasmids and BindingPCA pGJJ034 and pGJJ001 plasmids were previously described[23]. For library 4: pTB043 plasmid containing full-length SRC was described previously[51]. pTB043 is based on the same backbone as the AbundancePCA plasmids. The difference is that full-length SRC is fused to the DHFR[3] fragment at its N terminus and to the DHFR[1,2] fragment at its C terminus, so DHFR is reconstituted following correct folding of SRC, whereas unfolded SRC genotypes result in degradation of the fusion protein.

**Libraries construction.** Libraries 1–3: libraries were constructed in two steps. First, an IDT primer containing the chosen combination of mutations was assembled by Gibson into the mutagenesis plasmid pGJJ286. Libraries were then cloned into the yeast plasmids AbundancePCA pGJJ045 and BindingPCA pGJJ001 by digestion/ligation. For the first step, the libraries into the mutagenesis plasmid were assembled by Gibson reaction (in-house preparation) of two fragments. The vector fragment was obtained by polymerase chain reaction (PCR) amplification of pGJJ286 with the oligos shown in Supplementary Tables 1 and 2, incubated with DpnI to remove the template and gel purified using QIAquick gel extraction kit (Qiagen).

The insert fragment was obtained by mixing equimolar amounts of IDT mutation primer (Supplementary Tables 1 and 2) and a reverse elongation primer (Supplementary Tables 1 and 2) and incubating for one cycle of annealing/extension with Q5 polymerase (New England Biolabs). dsDNA product was then incubated with ExoSAP-IT (Applied Biosystems) to remove the remaining ssDNA and purified with MinElute columns (Qiagen). 100 ng of vector in a molar ratio of 1:5 with the insert was incubated for 3 h at 50 °C with a Gibson mix 2× prepared in-house. The reaction was desalted by dialysis with membrane filters (MF-Millipore) for 1 h and concentrated 4× using a SpeedVac concentrator (Thermo Scientific). DNA was then transformed into NEB 10-beta High Efficiency Electrocompetent *E. coli*. Cells were allowed to recover in SOC medium (NEB 10-beta Stable Outgrowth Medium) for 30 min and later transferred to LB medium with spectinomycin overnight. A fraction of cells was also plated into spectinomycin + LB + agar plates to estimate the total number of transformants. 100 ml of each saturated *E. coli* culture was collected the next morning to extract the mutagenesis plasmid library using the QIAfilter Plasmid Midi Kit (QIAGEN). To obtain the final libraries into the yeast plasmids, libraries in pGJJ286 plasmid were digested with NheI and HindIII, gel purified (MinElute Gel Extraction Kit, QIAGEN) and cloned into pGJJ045 or pGJJ034 digested plasmids with T4 ligase (New England Biolabs) by temperature-cycle ligation following the manufacturer's instructions, 67 fmol of backbone and 200 fmol of insert in a 33.3-µl reaction. The ligation was desalted by dialysis using membrane filters for 1 h, concentrated 4× using a SpeedVac concentrator (Thermo Scientific) and transformed into NEB 10-beta High Efficiency Electrocompetent *E. coli* cells.

Library 4: this library was constructed in one step by Gibson reaction of two fragments. The vector fragment was obtained by amplification of pTB043 plasmid with the oligos shown in the Supplementary Tables 1 and 2. The second fragment was obtained with ten cycles of PCR using mutated IDT primer as template (Supplementary Tables 1 and 2).

**Methotrexate yeast selection assay.** The yeast selection assay was previously described[23]. The high-efficiency yeast transformation protocol described below (adjusted to a pre-culture of 200 ml of YPDA) was scaled up or down, depending on the number of transformants for each library (Supplementary Table 2). Three independent pre-cultures of BY4742 were grown in 20 ml of standard YPDA at 30 °C overnight. The next morning, the cultures were diluted into 200 ml of pre-warmed YPDA at an $OD_{600nm} = 0.3$ and incubated at 30 °C for 4 h. Cells were then collected and centrifuged for 5 min at 3,000$g$, washed with sterile water and SORB medium, resuspended in 8.6 ml of SORB and incubated at room temperature for 30 min. After incubation, 175 µl of 10 mg ml$^{-1}$ boiled salmon sperm DNA (Agilent Genomics) and 3.5 µg of plasmid library were added to each tube of cells and mixed gently. 35 ml of plate mixture was added to each tube to be incubated at room temperature for a further 30 min. 3.5 ml of DMSO was added to each tube and the cells were then heat shocked at 42 °C for 20 min (inverting tubes from time to time to ensure homogenous heat transfer). After heat shock, cells were centrifuged and resuspended in approximately 50 ml of recovery media and allowed to recover for 1 h at 30 °C. Cells were then centrifuged, washed with SC-URA medium and resuspended in 200 ml SC-URA. 10 µl was plated on SC-URA Petri dishes and incubated for about 48 h at 30 °C to measure the transformation efficiency. The independent liquid cultures were grown at 30 °C for about 48 h until saturation. Saturated cells were diluted again to $OD_{600nm} = 0.1$ in SC-URA/MET/ADE media and allowed to grow four generations until $OD_{600nm} = 1.6$ at 30 °C and 200 rpm. A fraction of the culture was then used to inoculate 200 ml of competition media containing methotrexate at a starting $OD_{600nm} = 0.05$ and the rest was collected and pellets frozen and stored as input. Cells in competition media were allowed to grow for 3–5 generations (Supplementary Table 2), collected and frozen and stored as output.

**DNA extractions and plasmid quantification.** The DNA extraction protocol used was previously described[23]. The protocol below is for 100 ml of collected culture at $OD_{600nm} \approx 1.6$. Protocols were scaled up or down, depending on the library (Supplementary Table 2). Cell pellets (one for each experiment input/output replicate) were resuspended in 1 ml of DNA extraction buffer, frozen by dry ice/ethanol bath and incubated at 62 °C in a water bath twice. Subsequently, 1 ml of phenol/chloro/isoamyl in a ratio of 25:24:1 (equilibrated in 10 mM Tris-HCl, 1 mM EDTA, pH 8.0) was added, together with 1 g of acid-washed glass beads (Sigma Aldrich) and the samples were vortexed for 10 min. Samples were centrifuged at room temperature for 30 min at 4,000 rpm and the aqueous phase was transferred into new tubes. The same step was repeated twice. 0.1 ml of NaOAc 3 M and 2.2 ml of pre-chilled absolute ethanol were added to the aqueous phase. The samples were gently mixed and incubated at −20 °C for at least 30 min. After that, they were centrifuged for 30 min at full speed at 4 °C to precipitate the DNA. The ethanol was removed and the DNA pellet was allowed to dry overnight at room temperature. DNA pellets were resuspended in 0.6 ml TE 1X and treated with 5 µl of RNase A (10 mg ml, Thermo Scientific) for 30 min at 37 °C. To desalt and concentrate the DNA solutions, the QIAEX II Gel Extraction Kit was used (50 µl of QIAEX II beads, QIAGEN). The samples were washed twice with PE buffer and eluted twice by 125 µl of 10 mM Tris-HCl buffer, pH 8.5. Finally, plasmid concentrations in the total DNA extract (which also contained yeast genomic DNA) were quantified by quantitative PCR using the primer pair oGJJ152–oGJJ153 that binds to the ori region of the plasmids.

**Sequencing library preparation.** Libraries 1–3: this was shown in ref. 23. Briefly, the sequencing libraries were constructed in two consecutive PCR assays. The first PCR (PCR1) was designed to amplify the mutated protein of interest and to increase the nucleotide complexity of the first sequenced bases by introducing frame-shift bases between the adapters and the sequencing region of interest (Supplementary Tables 1 and 2). The second PCR (PCR2) was necessary to add the remainder of the Illumina adapter and demultiplexing indexes. PCR2 reactions were run for each sample independently using Hot Start High-Fidelity DNA Polymerase. In this second PCR, the remaining parts of the Illumina adapters were added to the library amplicon. The forward primer (5′ P5 Illumina adapter) was the same for all samples (GJJ_1J), whereas the reverse primer (3′ P7 Illumina adapter) differed by the barcode index (Supplementary Table 3) to be subsequently pooled and demultiplexed after deep sequencing. All samples were pooled in an equimolar ratio and gel purified using the QIAEX II Gel Extraction Kit. The purified amplicon library pools were subjected to Illumina 150-bp paired-end NextSeq500 sequencing at the CRG Genomics Core Facility.

Library 4: the method for preparing the library for sequencing was the same as for the other libraries but in the second PCR step, we used a barcoded index in the forward primer as well (5′ P5 Illumina adapter). The purified amplicon library pool was sequenced with an Illumina paired-end NextSeq2000 machine this time.

**Sequencing data processing**
FastQ files from paired-end sequencing of all AbundancePCR and BindingPCR experiments were processed with DiMSum v1.3 (ref. 52) using default settings with small adjustments (https://github.com/lehner-lab/DiMSum). Supplementary Table 4 contains DiMSum fitness estimates and associated errors for all experiments. Experimental design files and command-line options required for running DiMSum on these datasets are available on GitHub (https://github.com/lehner-lab/archstabms). Variants with fewer than ten input read counts in any replicate were discarded ('fitnessMinInputCountAll' option), that is, only variants observed in all replicates above this threshold were retained. For library 1, we also included fitness estimates that derived from a subset of replicates whose input read counts exceeded this threshold ('fitnessMinInputCountAny' option; see Fig. 1).

For library 1, we also included a wild-type-only sample for sequencing using pGJJ046 as template to derive empirical estimates of sequencing error rates. The FastQ file for this sample was processed identically to those of the replicate input/output samples in the first-pass analysis with DiMSum with permissive base quality thresholds ('vsearchMinQual = 5' and 'vsearchMaxee = 1000'). Read counts for all variants were then adjusted by subtracting the expected number of sequencing errors derived from the wild-type-only sample and proportional to the total sequencing library size of each sample. Finally, fitness estimates and associated errors for library 1 were then obtained from the resulting corrected variant counts with DiMSum ('countPath' option).

### Thermodynamic modelling with MoCHI

We used MoCHI[43] (https://github.com/lehner-lab/MoCHI) to fit all thermodynamic models to combinatorial DMS data using default settings with small adjustments. The software is based on our previously described genotype–phenotype modelling approach[23], with extra functionality and improvements for ease of use and flexibility[24,43]. Models fit to shallow (double-mutant) libraries and used in the analyses described in this work (for example, combinatorial mutagenesis library designs) were obtained using the original software implementation[23].

We model protein folding as an equilibrium between two states: unfolded (u) and folded (f), and protein binding as an equilibrium between three states: unfolded and unbound (uu), folded and unbound (fu) and folded and bound (fb). We assume that the probability of the unfolded and bound state (ub) is negligible and free energy changes of folding and binding are additive, that is, the total binding and folding free energy changes of an arbitrary variant relative to the wild-type sequence is simply the sum over residue-specific energies corresponding to all constituent single aa substitutions.

We configured MoCHI parameters to specify a neural network architecture consisting of additive trait layers (free energies) for each biophysical trait to be inferred (folding or folding and binding for AbundancePCA or BindingPCA, respectively), as well as one linear transformation layer per observed phenotype. The specified nonlinear transformations 'TwoStateFractionFolded' and 'ThreeStateFractionBound' derived from the Boltzmann distribution function relate energies to proportions of folded and bound molecules, respectively (see Figs. 2a and 4e,f). The target (output) data to fit the neural network comprise fitness scores for the wild-type and aa substitution variants of all mutation orders. The inclusion of both first-order and second-order (pairwise energetic coupling) model coefficients in the models was specified using the 'max_interaction_order' option.

A random 30% of aa substitution variants of all mutation orders was held out during model training, with 20% representing the validation data and 10% representing the test data. Validation data were used to evaluate training progress and optimize hyperparameters (batch size). Optimal hyperparameters were defined as those resulting in the smallest validation loss after 100 training epochs. Test data were used to assess final model performance.

MoCHI optimizes the parameters $\theta$ of the neural network using stochastic gradient descent on a loss function $\mathcal{L}[\theta]$ based on a weighted and regularized form of mean absolute error:

$$\mathcal{L}[\theta] = 1/N \sum_{n=0}^{N-1} |y_n - \hat{y}_n| \ \sigma_n^{-1} + \lambda_2 \ \|\theta\|^2$$

in which $y_n$ and $\sigma_n$ are the observed fitness score and associated standard error, respectively, for variant $n$, $\hat{y}_n$ is the predicted fitness score, $N$ is the batch size and $\lambda_2$ is the $L_2$ regularization penalty. To penalize very large free energy changes (typically associated with extreme fitness scores), we set $\lambda_2$ to $10^{-6}$, representing light regularization. The mean absolute error is weighted by the inverse of the fitness error ($\sigma_n^{-1}$) to downweight the contribution of less confidently estimated fitness scores to the loss. Furthermore, to capture the uncertainty in fitness

estimates, the training data were replaced with a random sample from the fitness error distribution of each variant. The validation and test data were left unaltered.

Models were trained with default settings, that is, for a maximum of 1,000 epochs using the Adam optimization algorithm with an initial learning rate of 0.05 (except for library 1, for which we used an initial learning rate of 0.005). MoCHI reduces the learning rate exponentially ($\gamma = 0.98$) if the validation loss has not improved in the most recent ten epochs compared with the preceding ten epochs. Also, MoCHI stops model training early if the wild-type free energy terms over the most recent ten epochs have stabilized (standard deviation $\leq 10^{-3}$).

Free energies are calculated directly from model parameters as follows: $\Delta G_b = \theta_b RT$ and $\Delta G_f = \theta_f RT$, in which $T = 303$ K and $R = 0.001987$ kcal K$^{-1}$ mol$^{-1}$. We estimated the confidence intervals of model-inferred free energies using a Monte Carlo simulation approach. The variability of inferred free energy changes was calculated between ten separate models fit using data from: (1) independent random training–validation–test splits and (2) independent random samples of fitness estimates from their underlying error distributions. Confident inferred free energy changes are defined as those with Monte Carlo simulation-derived 95% confidence intervals < 1 kcal mol$^{-1}$. Supplementary Table 5 contains inferred binding and folding free energy changes and energetic couplings from all second-order models.

### Linear model to predict energetic coupling strength

We built a linear model to predict energetic coupling strength (absolute value of energetic coupling terms) from 12 features (see Fig. 3e), comprising five distance metrics for residue pairs or positions thereof in the protein structure: backbone distance (linear 1D distance separating residue pairs along the primary aa sequence), inter-residue distance (minimal side-chain heavy-atom distance in 3D space), number of core residues (0, 1 or both residues in the pair with RSASA < 0.25), number of binding interface residues (0, 1 or both with minimal side-chain heavy-atom distance to the ligand < 5 Å), number of beta-sheet residues (0, 1 or both in beta strands) and seven features describing the number of chemical bonds or interactions between the atoms of pairs of residues as calculated using the GetContacts software tool (https://getcontacts.github.io/): backbone to backbone hydrogen bonds, side chain to backbone hydrogen bonds, side chain to side chain hydrogen bonds, pi–cation interactions, pi-stacking interactions, salt-bridge interactions and van der Waals interactions. Before running GetContacts, we used PyMOL to fill missing hydrogens ('h_add' command), FoldX[53] to restore the wild-type proline at position 54 that is mutated in the reference crystal structure (PDB: 2VWF; 'PositionScan' command) and removed GAB2 ligand atoms. The training dataset comprised energetic couplings inferred from library 1 and the test set comprised independently inferred energetic couplings from library 3 (see Fig. 3f).

### Reporting summary

Further information on research design is available in the Nature Portfolio Reporting Summary linked to this article.

### Data availability

All DNA sequencing data have been deposited in the Gene Expression Omnibus (GEO) with accession number GSE246322. Associated fitness measurements and free energies are provided in Supplementary Tables 4 and 5. Shallow double-mutant ddPCA DNA sequencing data for GRB2-SH3 and PSD95-PDZ3 are available in the GEO with accession number GSE184042, and the processed data used in this study can be found in Supplementary Tables 6 and 7 of the corresponding publication (https://doi.org/10.1038/s41586-022-04586-4). Protein structures are available from the Protein Data Bank for GRB2-SH3 (entry ID: 2VWF) and SRC (entry ID: 2SRC).

## Code availability

Source code for fitting thermodynamic models (MoCHI) is available at https://github.com/lehner-lab/MoCHI. Source code for all downstream analyses, including DiMSum and MoCHI configuration files, and to reproduce all figures described here is available at https://github.com/lehner-lab/archstabms. An archive of this repository is also publicly available on Zenodo at https://doi.org/10.5281/zenodo.11671164 (ref. 54).

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

**Acknowledgements** This work was funded by a European Research Council Advanced Grant (883742), the Spanish Ministry of Science and Innovation (LCF/PR/HR21/52410004, EMBL Partnership, Severo Ochoa Centre of Excellence), the Bettencourt Schueller Foundation, the AXA Research Fund, Agencia de Gestio d'Ajuts Universitaris i de Recerca (AGAUR, 2017 SGR 1322) and the CERCA Program/Generalitat de Catalunya. A.J.F. was funded by a Ramón y Cajal fellowship (RYC2021-033375-I) financed by the Spanish Ministry of Science and Innovation (MCIN/AEI/10.13039/501100011033) and the European Union (NextGenerationEU/PRTR). A.M.-A. was funded by a fellowship from "laCaixa" Foundation (ID 100010434, fellowship code B006052). We thank all members of the Lehner Lab for helpful discussions and suggestions.

**Author contributions** B.L. and A.J.F. conceived the project, motivated in part by preliminary unpublished analyses of evolutionary couplings performed by J.M.S. B.L., A.J.F. and A.M.-A. designed the combinatorial mutagenesis libraries and experiments. A.B. provided unpublished data and advice for the SRC experiments. A.M.-M. and C.H.-C. performed the experiments, with help from A.J.F. A.J.F. led the data analysis, with help from A.M.-A. A.J.F. and B.L. wrote the manuscript, with input from A.M.-A. and C.H.-C.

**Competing interests** A.J.F. and B.L. are founders, employees and shareholders of ALLOX. J.M.S. is a founder, employee and shareholder of factorize.bio.

**Additional information**
**Correspondence and requests for materials** should be addressed to Andre J. Faure or Ben Lehner.

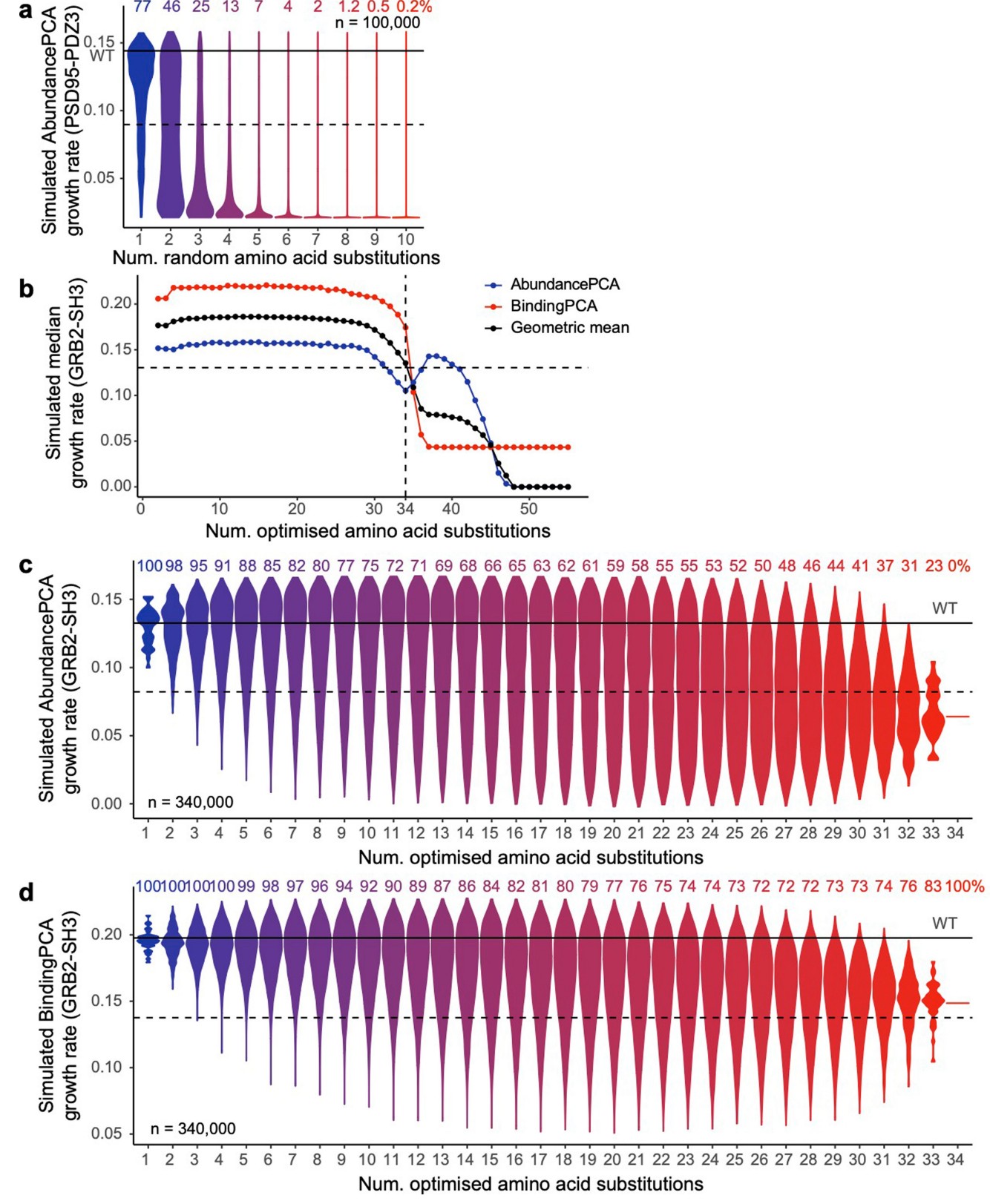

**Extended Data Fig. 1 | Combinatorial mutagenesis library 1 design and simulations. a**, Violin plot showing distributions of simulated AbundancePCA growth rates (assuming additivity of individual inferred folding free energy changes[23]) versus number of random aa substitutions ($n = 100,000$) for PSD95-PDZ3. Violins are scaled to have the same maximum width. **b**, Simulated median AbundancePCA/BindingPCA growth rates of optimal combinatorial libraries of increasing maximum aa Hamming distances from the wild type. The horizontal dashed line indicates the 70th percentile of the maximal geometric mean (black). The vertical dashed line indicates the number of aa substitutions selected ($n = 34$) for the synthesized combinatorial mutagenesis library 1. **c**, Violin plot showing simulated distributions of AbundancePCA growth rates versus number of aa substitutions for combinatorial mutagenesis library 1. **d**, Similar to panel **c** but showing simulated distributions for BindingPCA growth rates. In panels **c** and **d**, the percentage of folded and bound protein variants (predicted fraction folded or bound molecules > 0.5) is shown at each Hamming distance from the wild type.

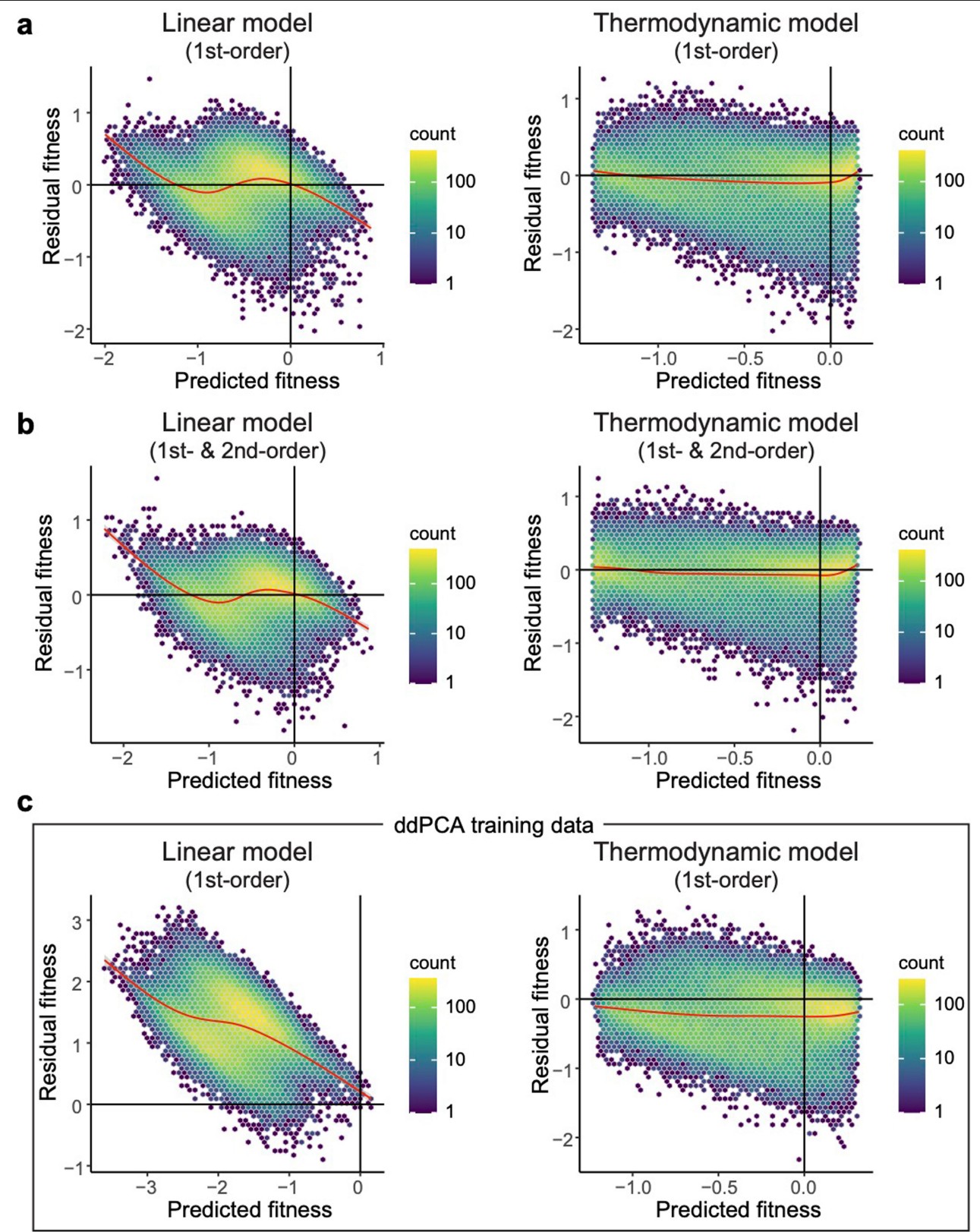

**Extended Data Fig. 2 | Residuals versus fitted values for linear and thermodynamic models fit to AbundancePCA data from combinatorial library 1. a**, Residual fitness (observed − predicted) versus predicted fitness for first-order linear models (left) and first-order thermodynamic models (right) evaluated on GRB2-SH3 combinatorial AbundancePCA data (combinatorial library 1; see Fig. 1). The smoothed conditional mean (generalized additive model) is shown in red. **b**, Similar to panel **a** except models include all first-order and second-order genetic interaction (energetic coupling) terms/coefficients. **c**, Similar to panel **a** except results are shown for models that were trained on GRB2-SH3 ddPCA data consisting of single and double aa substitutions only[23].

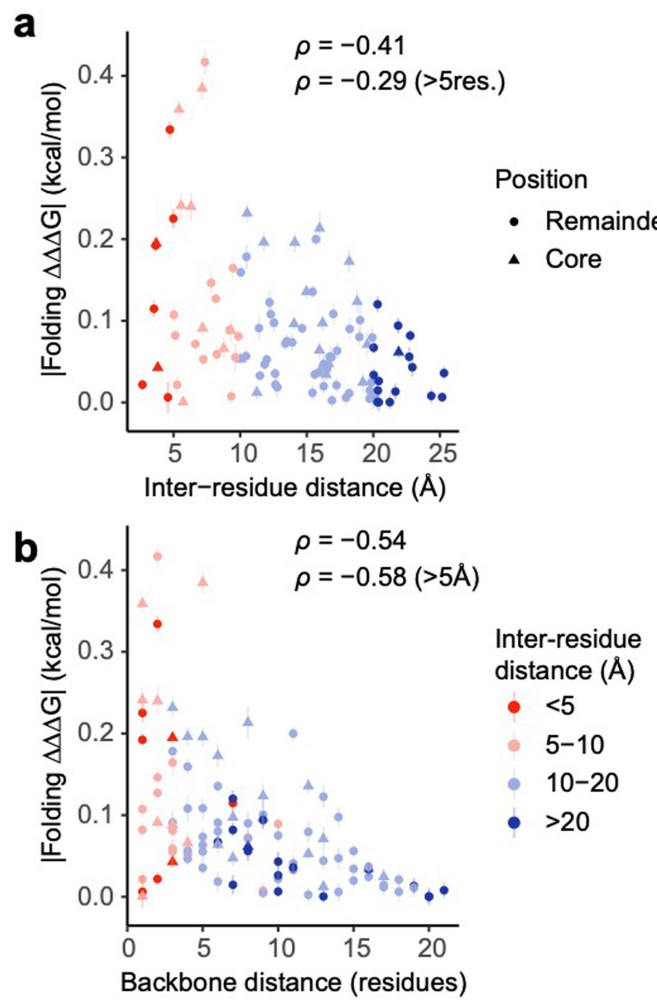

**Extended Data Fig. 3 | Structural determinants of energetic couplings inferred from ddPCA data from combinatorial library 3. a**, Relationship between folding coupling energy strength and minimal inter-residue side-chain heavy-atom distance for combinatorial library 3 (see Fig. 4). The mean is shown and error bars indicate 95% confidence intervals from a Monte Carlo simulation approach ($n = 10$ experiments). Points are coloured by binned inter-residue distances (see legend in panel **b**). Spearman's $\rho$ is shown for all couplings, as well as those involving pairs of residues separated by more than five residues in the primary sequence. Core residues are indicated as triangles. **b**, Relationship between folding coupling energy strength and linear sequence (backbone) distance in number of residues. The measure of centre and error bars are as defined in panel **a**.

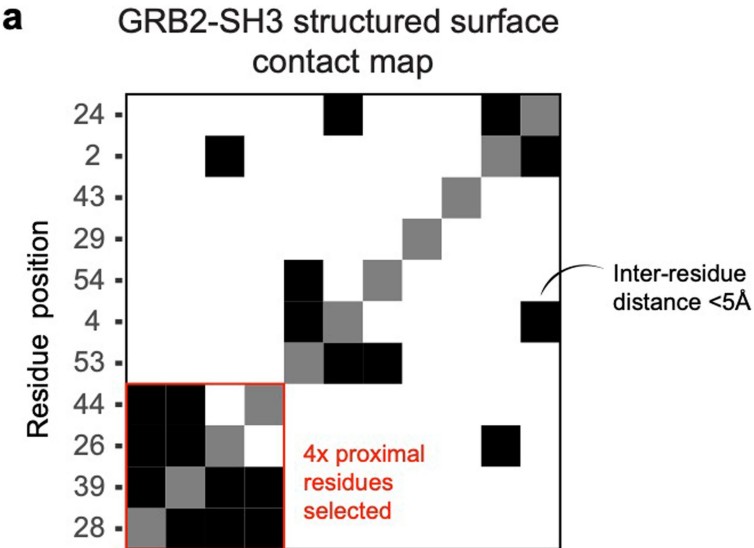

**a** GRB2-SH3 structured surface contact map

Inter-residue distance <5Å

4x proximal residues selected

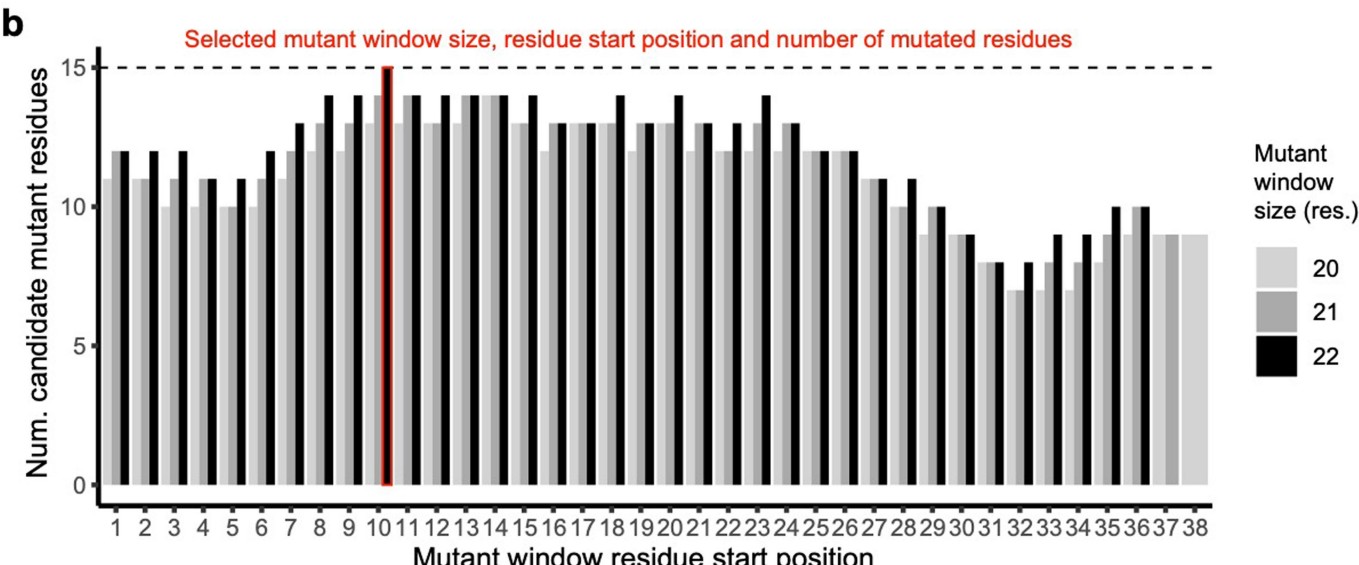

**b** Selected mutant window size, residue start position and number of mutated residues

Mutant window size (res.)
- 20
- 21
- 22

Num. candidate mutant residues

Mutant window residue start position

**Extended Data Fig. 4 | Design of combinatorial mutagenesis libraries 2 and 3. a**, Clustered heat map showing structural contacts (minimal side-chain heavy-atom distance < 5 Å) between all GRB2-SH3 surface residues (RSASA ≥ 0.25) existing in secondary structure elements. The four highlighted residues are all physically proximal and were selected as the targets for library 2 saturation combinatorial mutagenesis (see Extended Data Fig. 5). **b**, Bar plot indicating the number of candidate mutant residues in stretches of 20, 21 and 22 consecutive residues in GRB2-SH3 used to design mutagenesis library 3. Candidate mutations were defined as single aa substitutions with mild effects (within one-third of the AbundancePCA fitness interquartile range of the wild type[23]) in close proximity in the primary sequence and reachable by single-nucleotide substitutions while avoiding mutations in binding interface residues (minimal side-chain heavy-atom distance to the ligand < 5 Å). The selected mutant window size (22 aa residues), residue start position (10) and number of mutated residues (15) is indicated. The final library consisted of all combinations of the following randomly selected candidate mutations at these 15 positions: D10N, P11A, D14N, G15E, G18C, R20S, R21Q, D23E, F24I, H26L, V27I, M28K, D29E, N30T and S31T (see Fig. 4).

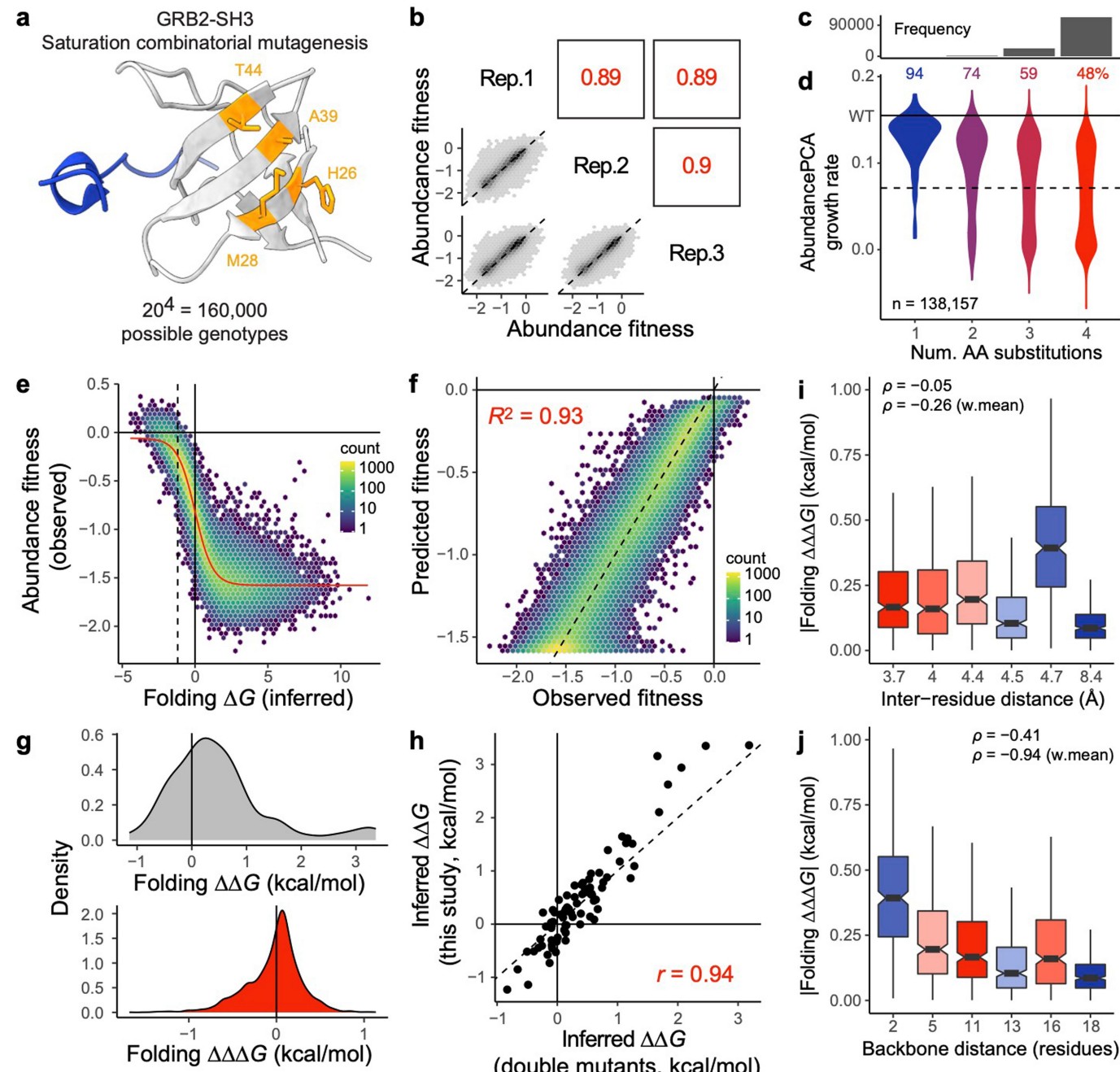

**Extended Data Fig. 5 | Saturation combinatorial mutagenesis of a protein surface patch. a**, 3D structure of GRB2-SH3 (PDB: 2VWF) indicating four residues targeted for saturation combinatorial mutagenesis (orange, library 2) and GAB2 ligand (blue). See also Extended Data Fig. 4. **b**, Scatter plots showing the reproducibility of fitness estimates from triplicate AbundancePCA experiments. Pearson's *r* indicated in red. Rep., biological replicate. **c**, Histogram showing the number of observed aa variants at increasing Hamming distances from the wild type, in which the *x* axis is shared with panel **d. d**, Violin plot showing distributions of AbundancePCA growth rates inferred from deep sequencing data versus number of aa substitutions. The percentage of folded protein variants (predicted fraction folded molecules > 0.5) is shown at each Hamming distance from the wild type. **e**, Nonlinear relationship (global epistasis) between observed AbundancePCA fitness and changes in free energy of folding. Thermodynamic model fit shown in red. **f**, Performance of energy model including all first-order and second-order genetic interaction (energetic coupling) terms/coefficients. **g**, Distributions of folding free energy changes (ΔΔG, grey) and pairwise energetic couplings (ΔΔΔG, red). **h**, Comparisons of the model-inferred single aa substitution free energy changes to previously reported estimates using GRB2-SH3 ddPCA data[23]. Pearson's *r* is shown. **i**, Box plots showing relationship between folding coupling energy strength and minimal inter-residue side-chain heavy-atom distance. Boxes are coloured by inter-residue distance. Spearman's *ρ* is shown for all couplings (*n* = 2,166 second-order coefficients), as well as the weighted mean per residue pair (*n* = 6 residue pairs). **j**, Relationship between folding coupling energy strength and linear sequence (backbone) distance in number of residues. Boxes are coloured as in panel **i**. For box plots in panels **i** and **j**: centre line, median; box limits, upper and lower quartiles; whiskers, 1.5× interquartile range; *n* = 2,166 second-order coefficients.

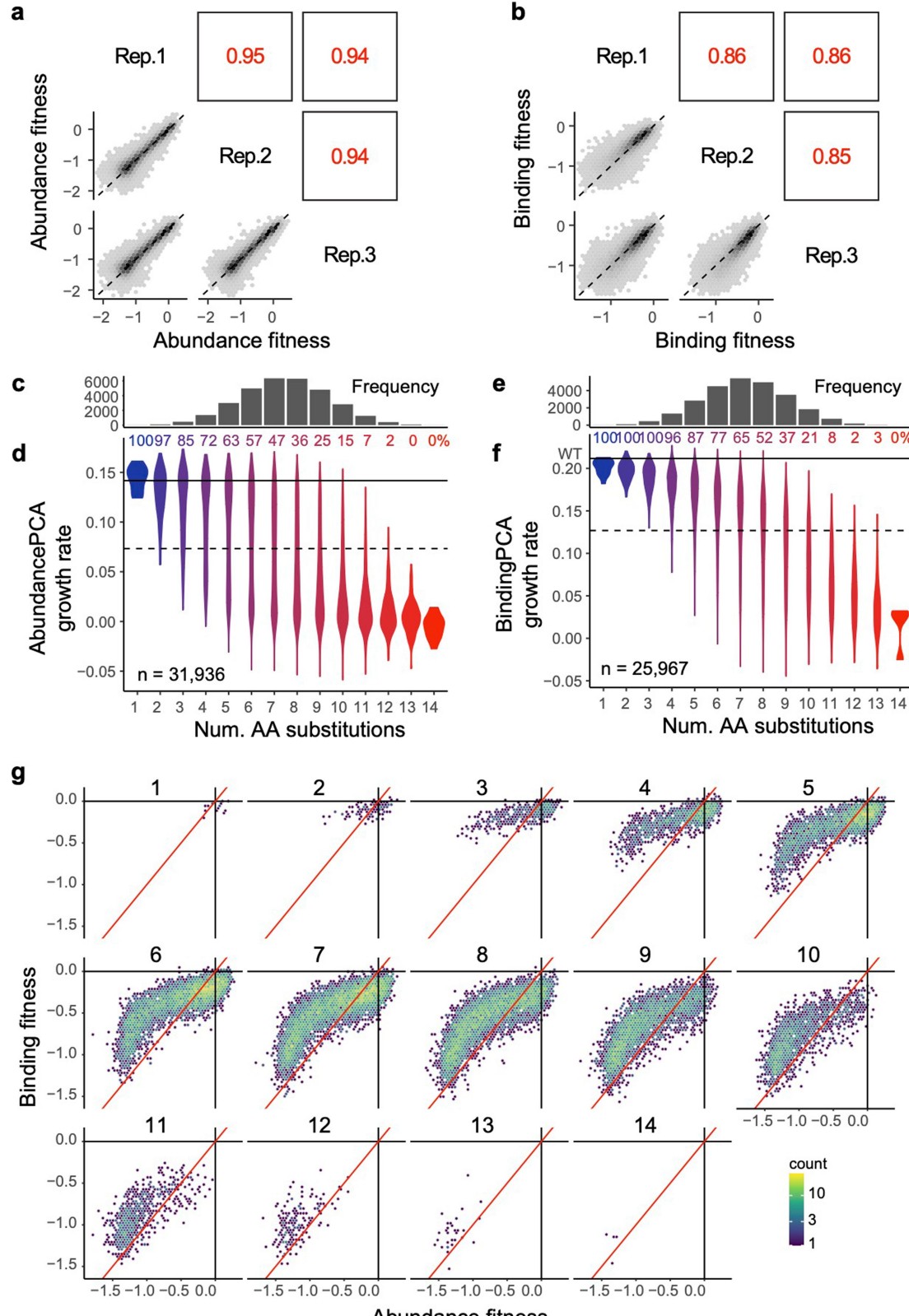

**Extended Data Fig. 6** | See next page for caption.

**Extended Data Fig. 6 | ddPCA data from combinatorial library 3 show that abundant multi-mutants are binding-competent (have conserved fold).** **a**, Scatter plots showing the reproducibility of fitness estimates from triplicate AbundancePCA experiments for combinatorial library 3 (see Fig. 4). Pearson's *r* indicated in red. Rep., biological replicate. **b**, Similar to panel **a** but showing results from triplicate BindingPCA experiments (same as Fig. 4c). **c**, Histogram showing the number of observed aa variants at increasing Hamming distances from the wild type for AbundancePCA, in which the *x* axis is shared with panel **d**. **d**, Violin plot showing distributions of AbundancePCA growth rates inferred from deep sequencing data versus number of aa substitutions. The percentage of folded protein variants (predicted fraction folded molecules > 0.5) is shown at each Hamming distance from the wild type. **e**,**f**, Similar to panels **c** and **d** but showing results for BindingPCA. The percentage of bound protein variants (predicted fraction folded molecules > 0.5) is shown at each Hamming distance from the wild type in panel **f**. **g**, 2D density plots comparing abundance and binding fitness for increasing Hamming distances 1–14 from the wild type as indicated.

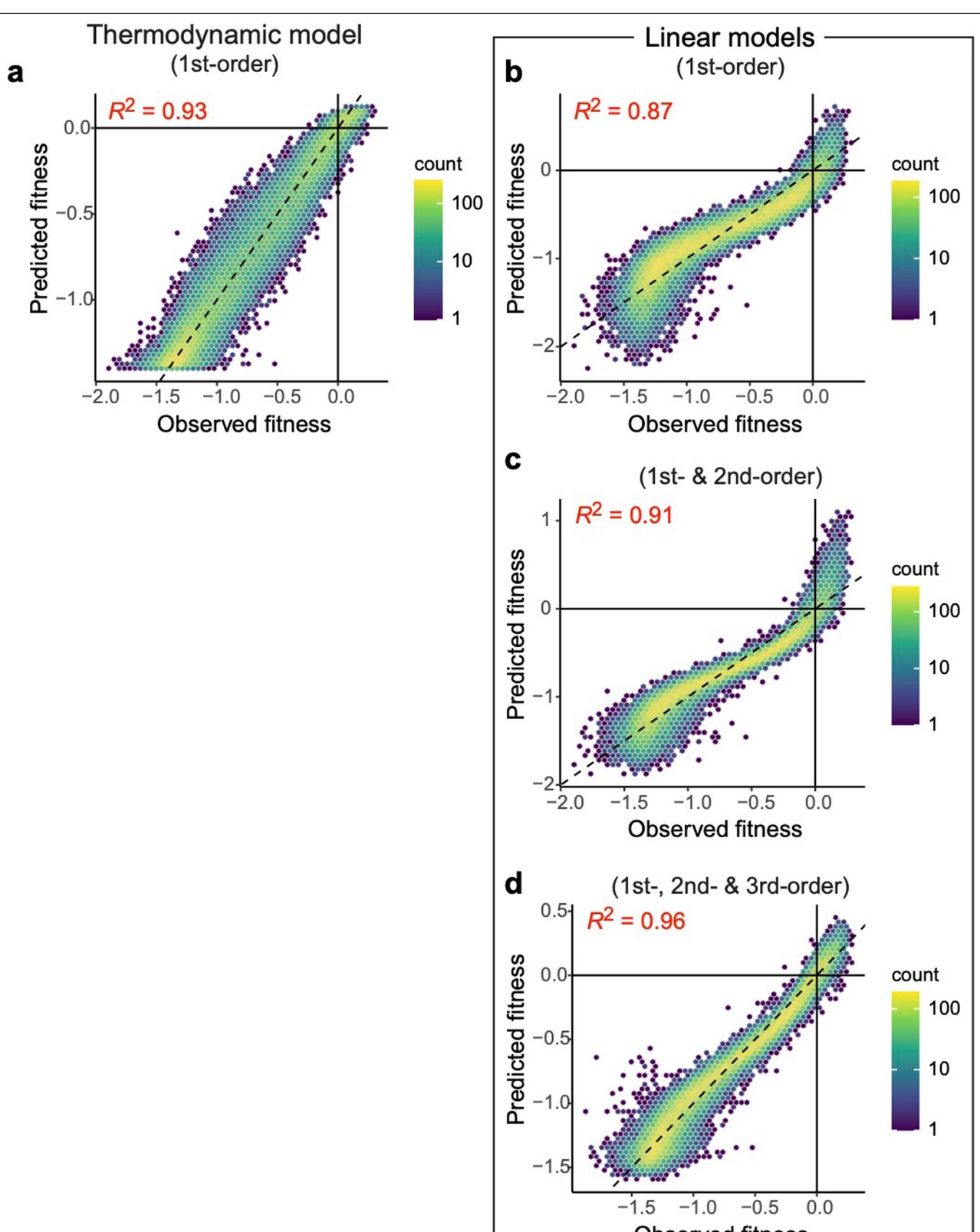

**Extended Data Fig. 7 | Performance of models fit to AbundancePCA data from combinatorial library 3. a**, Performance of first-order two-state thermodynamic model (folded and unfolded states) fit to AbundancePCA data from combinatorial library 3 (see Fig. 4). **b**–**d**, Performance of first-order (**b**), second-order (**c**) and third-order (**d**) linear models fit to AbundancePCA data from combinatorial library 3 (see Fig. 4). $R^2$ is the proportion of variance explained.

# Reporting Summary

## Statistics

For all statistical analyses, confirm that the following items are present in the figure legend, table legend, main text, or Methods section.

| n/a | Confirmed | |
|---|---|---|
| ☐ | ☒ | The exact sample size (*n*) for each experimental group/condition, given as a discrete number and unit of measurement |
| ☐ | ☒ | A statement on whether measurements were taken from distinct samples or whether the same sample was measured repeatedly |
| ☐ | ☒ | The statistical test(s) used AND whether they are one- or two-sided *Only common tests should be described solely by name; describe more complex techniques in the Methods section.* |
| ☒ | ☐ | A description of all covariates tested |
| ☐ | ☒ | A description of any assumptions or corrections, such as tests of normality and adjustment for multiple comparisons |
| ☐ | ☒ | A full description of the statistical parameters including central tendency (e.g. means) or other basic estimates (e.g. regression coefficient) AND variation (e.g. standard deviation) or associated estimates of uncertainty (e.g. confidence intervals) |
| ☐ | ☒ | For null hypothesis testing, the test statistic (e.g. *F*, *t*, *r*) with confidence intervals, effect sizes, degrees of freedom and *P* value noted *Give P values as exact values whenever suitable.* |
| ☒ | ☐ | For Bayesian analysis, information on the choice of priors and Markov chain Monte Carlo settings |
| ☒ | ☐ | For hierarchical and complex designs, identification of the appropriate level for tests and full reporting of outcomes |
| ☐ | ☒ | Estimates of effect sizes (e.g. Cohen's *d*, Pearson's *r*), indicating how they were calculated |

*Our web collection on statistics for biologists contains articles on many of the points above.*

## Software and code

Policy information about availability of computer code

| Data collection | FastQ files from paired-end sequencing of all bindingPCA and abundancePCA experiments were processed with DiMSum v1.3 using default settings with minor adjustments: https://github.com/lehner-lab/DiMSum. All experimental design files and bash scripts with command-line options required for running DiMSum on these datasets are available at https://github.com/lehner-lab/archstabms. |
|---|---|
| Data analysis | Source code for fitting thermodynamic models (MoCHI v0.9) is available at https://github.com/lehner-lab/MoCHI. Source code for all downstream analyses, including DiMSum and MoCHI configuration files and to reproduce all figures described here is available at https://github.com/lehner-lab/archstabms. An archive of this repository is also publicly available on Zenodo at https://zenodo.org/doi/10.5281/zenodo.11671164. Chemical bonds or interactions were calculated using the GetContacts software package (https://github.com/getcontacts/getcontacts) with version status: July 25th 2023. PyMOL v2.5.2 was used to fill missing hydrogens. FoldX v4 was used to restore the wild-type Proline at position 54 that is mutated in the reference crystal structure (PDB: 2VWF, "PositionScan" command). |

For manuscripts utilizing custom algorithms or software that are central to the research but not yet described in published literature, software must be made available to editors and reviewers. We strongly encourage code deposition in a community repository (e.g. GitHub). See the Nature Portfolio guidelines for submitting code & software for further information.

## Data

Policy information about availability of data

All manuscripts must include a data availability statement. This statement should provide the following information, where applicable:
- Accession codes, unique identifiers, or web links for publicly available datasets
- A description of any restrictions on data availability
- For clinical datasets or third party data, please ensure that the statement adheres to our policy

All DNA sequencing data have been deposited in the Gene Expression Omnibus (GEO) with accession number GSE246322: https://www.ncbi.nlm.nih.gov/geo/query/acc.cgi?acc=GSE246322. Associated fitness measurements and free energies are provided in Supplementary tables 4 and 5. Shallow double mutant ddPCA DNA sequencing data for GRB2-SH3 and PSD95-PDZ3 is available in GEO with accession number GSE184042: https://www.ncbi.nlm.nih.gov/geo/query/acc.cgi?acc=GSE184042 and the processed data used in this study can be found in Supplementary Tables 6 and 7 of the corresponding publication (DOI: 10.1038/s41586-022-04586-4). Protein structures for GRB2-SH3 (Entry ID: 2VWF) and SRC (Entry ID: 2SRC) are available from the Protein Data Bank (PDB): https://www.rcsb.org/

## Research involving human participants, their data, or biological material

Policy information about studies with human participants or human data. See also policy information about sex, gender (identity/presentation), and sexual orientation and race, ethnicity and racism.

| | |
|---|---|
| Reporting on sex and gender | N/A |
| Reporting on race, ethnicity, or other socially relevant groupings | N/A |
| Population characteristics | N/A |
| Recruitment | N/A |
| Ethics oversight | N/A |

Note that full information on the approval of the study protocol must also be provided in the manuscript.

# Field-specific reporting

Please select the one below that is the best fit for your research. If you are not sure, read the appropriate sections before making your selection.

☒ Life sciences ☐ Behavioural & social sciences ☐ Ecological, evolutionary & environmental sciences

For a reference copy of the document with all sections, see nature.com/documents/nr-reporting-summary-flat.pdf

# Life sciences study design

All studies must disclose on these points even when the disclosure is negative.

| | |
|---|---|
| Sample size | Sample sizes during the construction of the mutant libraries and the yeast competition experiments were always several fold larger than the bottlenecked library size to ensure losing as few amino acid variants as possible during the experiments. The minimum number of yeast transformants in each of the bulk competition replicates (the strongest bottleneck in the experimental design) was calculated so that mutations in the library would be found on average in 20-25 different cells. |
| Data exclusions | Sequencing reads that did not pass the QC filters using DiMSum v1.3 (https://github.com/lehner-lab/DiMSum) were excluded. For mutagenesis library 1 read counts for all variants were adjusted by subtracting the expected number of sequencing errors derived from the wild-type-only sample and proportional to the total sequencing library size of each sample. |
| Replication | All bulk yeast competitions per assay and protein library were performed in triplicates. All attempts of replication were successful. |
| Randomization | Not relevant for this study because mutant libraries were created systematically and screened in bulk selections experiments. |
| Blinding | Not relevant for this study because mutant libraries were created systematically and screened in bulk selections experiments. |

# Reporting for specific materials, systems and methods

We require information from authors about some types of materials, experimental systems and methods used in many studies. Here, indicate whether each material, system or method listed is relevant to your study. If you are not sure if a list item applies to your research, read the appropriate section before selecting a response.

## Materials & experimental systems

| n/a | Involved in the study |
|-----|----------------------|
| ☒ ☐ | Antibodies |
| ☐ ☒ | Eukaryotic cell lines |
| ☒ ☐ | Palaeontology and archaeology |
| ☒ ☐ | Animals and other organisms |
| ☒ ☐ | Clinical data |
| ☒ ☐ | Dual use research of concern |
| ☒ ☐ | Plants |

## Methods

| n/a | Involved in the study |
|-----|----------------------|
| ☒ ☐ | ChIP-seq |
| ☒ ☐ | Flow cytometry |
| ☒ ☐ | MRI-based neuroimaging |

## Eukaryotic cell lines

Policy information about cell lines and Sex and Gender in Research

| | |
|---|---|
| Cell line source(s) | Saccharomyces cerevisiae BY4742 (MATα his3Δ1 leu2Δ0 lys2Δ0 ura3Δ0) |
| Authentication | The cell line was not authenticated |
| Mycoplasma contamination | Not tested for Mycoplasma (not applicable) |
| Commonly misidentified lines (See ICLAC register) | *Name any commonly misidentified cell lines used in the study and provide a rationale for their use.* |

## Plants

| | |
|---|---|
| Seed stocks | *Report on the source of all seed stocks or other plant material used. If applicable, state the seed stock centre and catalogue number. If plant specimens were collected from the field, describe the collection location, date and sampling procedures.* |
| Novel plant genotypes | *Describe the methods by which all novel plant genotypes were produced. This includes those generated by transgenic approaches, gene editing, chemical/radiation-based mutagenesis and hybridization. For transgenic lines, describe the transformation method, the number of independent lines analyzed and the generation upon which experiments were performed. For gene-edited lines, describe the editor used, the endogenous sequence targeted for editing, the targeting guide RNA sequence (if applicable) and how the editor was applied.* |
| Authentication | *Describe any authentication procedures for each seed stock used or novel genotype generated. Describe any experiments used to assess the effect of a mutation and, where applicable, how potential secondary effects (e.g. second site T-DNA insertions, mosiacism, off-target gene editing) were examined.* |

