## [Peer Review file · Nature]

Manuscript Title: The genetic architecture of protein stability

Reviewer Comments & Author Rebuttals

Reviewer Reports on the Initial Version:

Referee #1:

Understanding how genotypes give rise to molecular and organismal phenotypes is arguably the central question of biology. Recent applications of machine learning to the problem have made major advances but may have limited interpretability. New sequencing-based high throughput genotype-phenotype assays have provided a wealth of datasets for probing this question as well. To date, these datasets have consisted of single substitutions, however, with a small number of double or higher-order combinations.

In this manuscript, Faure et al. measure the effects of combinatorial sets of mutations in several higher-order mutational scans to investigate how far biophysical models can be used to understand genotype-phenotype links. To avoid the combinatorial explosion in size that would result, the authors employ a clever and highly innovative 'Greedy' approach to design their library that harnesses their previous single substitution libraries (Faure, 2022, Nature) with the aim of maximizing the sampling of folded and functional mutants in their library. This approach is of significance and impact as it could help with efficient sampling of higher order combinatorial space.

One major open question they tackle is the nature of epistasis in these maps, i.e. non-additive free-energy changes in combinations of mutations, and particularly their sparsity (or how important progressively higher-order epistatic contributions are). Another open question is the complexity of these maps, and whether simple biophysical models are sufficient, or if more complex ML approaches are required, with the associated loss of direct mechanistically interpretable parameters. The authors, thorough analysis of their datasets, provide important answers to these questions, although the choice of their model system as a small protein domain limits the generalizability of their results to larger proteins with multiple folds.

This is a well-written, rigorous, thorough piece of work and will be of great interest to biologists broadly, with particular importance for protein biology, computational modeling, and evolutionary biochemistry communities. We feel that the argument and stakes could be sharpened, and there are some overstatements and important additional analyses that should be done. We recommend another round of revision at Nature after additional analyses are done.

Major Comments

1. The authors emphasize they screen a mutational space of $\sim 10^{10}$, whereas prior work has only reached $\sim 10^6$. This is somewhat misleading. The authors do sample from a large design space, however they are still experimentally measuring $\sim 10^5$ variants. This is, as they admit, a small fraction of the overall space. They argue that the vast majority of mutations will produce a misfolded protein, which is likely valid, and so the space of informative genotypes is actually much smaller than 10^{10} . To make this claim, however, they use the predicted additive folding energies from their prior single substitution experiment. This seems to beg the question, in that they are assuming that the additivity of single variants is sufficient to predict gross higher-order phenotypes in order to avoid comprehensively measuring higher-order phenotypes, which they would do to prove that additivity is sufficient to predict higher-order phenotypes. Also, one could argue that approaches like error-prone PCR similarly sparsely sample extremely large sequence spaces. We recommend some discussion clarifying how the sampling strategy differs from other approaches, and walking back the claim that they are comprehensively sampling a mutational space of 10^{10} .

2. The authors state throughout the results and in the conclusions that the data supports the notion that genetic landscapes and interactions are generally simple. We believe this claim requires more work. Although several obvious definitions for simplicity might apply, the authors do not explicitly define what they mean, so it is difficult to quantify the results here. Instead, they argue for simplicity by showing improved performance on prediction tasks using biophysical models with first or second-order interaction terms only. The saturation combinatorial mutational scan and the other higher order library both indeed show strong correlations with first- and second-order models (R^2 0.93 and 0.97). The Greedy dataset, however, has much lower correlation, giving an R^2 of 0.72 for the second-order model. Given that these fits apply to the same measurements on the same protein, the differences in correlation must have to do with the architecture of the different sequence spaces. The Greedy design, then, seems to create a space with increased importance of higher-order interactions, and so decreased simplicity in the terms of the authors. We recommend the authors improve their discussion of this implication, as well as more clearly state how “simplicity” is being determined. This may also help demonstrate the novelty of their technique, as the fact that most residues are not coupled has been described before going back to foundational work in combinatorial mutagenesis (Wells, 1990, *Biochemistry*, for example).

3. The greedy mutational scanning approach is highly novel and is the first of its kind that explores much higher-order mutational space beyond 2nd-order. However, the authors do not explore combinatorial trajectories in depth beyond 2nd order effects - there should be further analysis and likely additional figure panels that explore effects beyond those that are 2nd order. Otherwise it is unclear why such a greedy deep mutational scan was done and why these analyses couldn't have been done on their existing data from Faure 2022 or why a more comprehensive double mutational scan couldn't have been done as has been previously done on GB2 (Olson, CA, 2014). In particular there may be interesting mutational trajectories that could be explored with this data if one contrasts the results with natural predicted evolutionary trajectories within SH3 orthologs or paralogs using similar analysis as has been done for enzymes with lower throughput combinatorial mutational experiments (Weinreich, 2006, *Science*). While this specific analysis is not necessary - it would be good to see further analyses of the higher order space.

4. In Fig. 3 the authors train models on structural features to predict mutational effects on a combinatorial dataset inferred using a 1st and 2nd order biophysical model built on the Greedy dataset. That the authors get a model based solely on protein features that performs about as well as Rosetta/Foldx approaches suggests something may be learned here about what determines epistasis. The authors find that backbone distance is important for predicting the effects of mutations whereas inter-residue distance is not. In other words, the placement in the primary sequence seems to be important while the 3D contacts do not. The authors refer to this observation a number of times throughout the manuscript including the following statement:

‘The observation that mutation effects can be modulated by the sequence context at distal sites and that the strength of this coupling depends in part on the number of intervening residues – but not 3D distance – suggests that energetic perturbations are particularly efficiently transmitted over long distances along the covalent bond structure of the peptide backbone.’

This is a very surprising result, and seems to contradict the typical understanding of what determines the folding of proteins. We therefore believe it requires further quantification and discussion. In SH3 domains, circular permutation, where a protein’s N- and C-termini are swapped (which would change position in primary sequence but potentially maintain tertiary sequence), has been successfully performed (Viguera, 1996, *Nat. Struct. Biol.*). This appears to conflict with the broadest claims of the authors.

We also note that the features the authors include in the model may introduce structural information: information about pairwise residue interactions is encoded in their physicochemical interactions, which do provide informative features here. It is also possible that the inter-residue distance metric is not appropriate to describing the effects of a mutation, as the minimal heavy atom distance will change in mutant structures especially in the context of combinatorial mutations, which are the backgrounds that epistatic couplings would be occurring in.

Overall, that 3D contacts do not determine stability and backbone propagations is somewhat unorthodox and counterintuitive - this claim should be further theorized. An alternative helpful comparison that would be useful to contextualize the performance of the model in this manuscript would be to use structure-based effect predictors Rosetta/Foldx/Foldetta which will account for the structural perturbations of mutations and which could give a more reliable inter-residue distance as well (Gerasimavicius, *Protein Science*, 2023). Perhaps such a comparison would also be useful in guiding what features to include that are missing from the modeling currently - or in other words what the missing ‘0.5’ predicted folding $\Delta\Delta G$ is. Currently the model’s performance does not support the statement within the conclusion that the ‘genetic architecture of proteins is predictable and understandable’.

5. Epistasis and interactions between residues within proteins are largely determined by the physicochemistry of residues. However, there is very little discussion currently that considers how physicochemistry of residues changes behavior of interacting residues and presence of non-additivity. For example, aromatic and non-polar burial is classically thought to drive folding - is there a core aromatic in SH3? If so, what is the effect of removing the aromatic and replacing it elsewhere in the core? Similarly, ion interactions and charge swaps are the classic example for non-additivity -

is this seen within the dataset? How does changing the distance between ionic pairs or locations in the protein affect folding/function? The richness of this dataset calls out for a specific physicochemical analysis.

Minor Comments

1. Please add page and line numbers to help reviewers make specific references.
2. The authors provide an excellent repository for the analysis and we would like to point that out. We would recommend they use Zenodo or a similar service to provide a reference to a specific release version, however.
3. Page 3, paragraph 2: The authors cite Poelwijk (2019) and Sarkisyan (2016) for the claim that “combining random mutations in even moderate numbers nearly always results in unfolded proteins,” but these references don’t support that specific claim. Both references measure fluorescence phenotypes, rather than soluble or folded fractions, and so this claim should be adjusted.
4. Page 3, paragraph 2: The authors should clarify that by “expected to be folded” they mean when assuming additivity, based on their previous single-site scans.
5. Page 4, paragraph 4: The authors mention a “bottlenecked library,” but there are no other references to this.
6. Page 10, paragraph 3: The authors mention testing the model on an ‘independent combinatorial’ experiment. What exactly is this dataset? Is this from the same research group and assay? Is it also a small protein?
7. Fig. 2f (and discussion thereof): These densities approximate the distributions of fitness effects of the subsampled combinatorial space, which was designed with the assumption of additivity. Do these densities fairly represent the entire space? Is it surprising that the epistatic couplings have mean 0?
8. Fig. 3a,b: What is the relationship between primary and tertiary sequence in SH3? Interpreting the difference in correlations between inter-residue distance and $\Delta\Delta\Delta G$ vs backbone distance and $\Delta\Delta\Delta G$ would be helped by understanding how inter-residue distance and backbone distance are themselves correlated.
9. Fig. 4i: The closest class of residues (i.e., 2 residues apart) appears to have significantly larger effects on folding $\Delta\Delta\Delta G$ than all others. We wonder how they are driving the correlation - what would the spearman rho be if they were excluded?
10. It could be interesting to discuss whether different types of higher order libraries contain more or less folded proteins specifically. For instance, is the Greedy approach more or less enriched for folded proteins? Are there types of bootstrapping that could be used to more efficiently sample

higher order space?

11. Adding a short definition of “genetic architecture” would be useful for non-specialists.

12. We would also suggest a bit more care with the language of “energetic perturbations” being “transmitted over long distances along the covalent bond structure of the peptide backbone,” and similar – this, to me, suggests some physical mechanism or interaction (a causation) that is not necessarily being claimed (there is a clear correlation, however).

13. Additional care to the use of “generalizability” and similar language should be taken. A careless reader might interpret this to mean generalizability across proteins, whereas here it means generalizability across orders of mutations.

14. There are a few important citations that could help contextualize this for the reader. On the additivity of mutational effects, Wells (1990, *Biochemistry*) is an important foundational reference. The work of Michael Harms, and particularly his group’s approach to linearizing genotype-phenotype maps, should be addressed. Dryden, et al. (2008, *J. R. Soc. Interface*) is an important work for putting sequence space sizes into perspective as well.

Referee #2:

This paper explores the determinants of folding and binding for an SH3 domain of the protein GRB2. The authors experimentally measure folding and binding for several large libraries of combinatorial mutants. They then use their previously developed approach to infer the individual effects of single mutants as well as pairwise couplings between residues. The authors find that relatively sparse models (single and pairwise terms only) can model the overall fitness landscape very well, even in sequences far away from wild-type (>15 mutations away). This is noteworthy, but somewhat unsurprising: as the authors note, similar studies have come to similar conclusions (paragraph 2 of discussion). However, the number of mutations explored is much larger in this study than in previous work.

The main strengths of the paper are:

1. The paper explores a very large combinatorial sequence space (simultaneous mutations at 34 positions for folding and 14 positions for binding). This scale is exciting and novel, especially in the context of folding stability (most large-scale studies examine a functional phenotype such as binding or fluorescence).
2. The paper simultaneously infers coupling effects for both folding and binding on a more limited library scale (mutations at 14 sites). As the authors note, I think this is the first example of inferring couplings for both phenotypes on a large scale. These results add to our understanding of sequence-phenotype relationships and also provide valuable data for testing machine learning models.
3. The finding that folding couplings more strongly correlate with primary sequence separation rather than structural separation is surprising and noteworthy.
4. The paper is very clearly written with figures that are mostly easy to understand (with some exceptions below).

Overall the conclusions of the study are well supported by their data (with minor exceptions noted below). I think the study will be exciting and interesting to protein biophysicists and readers interested in the high-throughput technology and machine learning in protein science.

The main weaknesses of the paper are:

1. The abstract, introduction, and discussion contain many irrelevant (and arguably incorrect) claims about deep neural networks. It does not make sense to me to compare the number of parameters in a neural network to the number of epistatic terms required to model one protein's fitness landscape. Analogously, the human brain is a very complex neural network, but can still solve very simple math problems - this doesn't mean the brain is "too complicated". I would suggest removing all the discussion about neural networks, or at minimum revising the claims to be more specific and defensible (see below).

2. The abstract states that the authors "experimentally explore sequence spaces $>10^{10}$ ". This is misleading because it suggests that 10^{10} sequences were actually experimentally examined, when the actual number is 10^5 . I think the exploration of mutations at 34 sites is noteworthy and important, and perhaps this is a more accurate way of describing what was performed. Or perhaps "By experimentally examining 100,000 sequences out of a possible space of 10^{10} ..."

3. It is somewhat unclear how "physical" the authors model is. They refer to it as "physical" many times, and the parameters of their model are interpretable. However, we would expect truly physical models to be transferrable from one protein to another. In contrast, the author's model is phenomenological and only valid for this one protein. Perhaps different terminology (or a clearer definition of what makes the model physical) would be beneficial.

Other clarifications and suggestions to the author:

- Abstract: "However, these models are extremely complicated and provide little insight into the fundamental genetic architecture of proteins." - The amount of insight that can be gained from deep neural networks is very contentious and the "little insight" claim in the abstract is very debatable. A less contestable claim would be that individual parameters in neural networks don't have physical interpretations (but model predictions do have physical interpretations).

- Abstract: "Here, by experimentally exploring sequence spaces $>10^{10}$ " - As above, this is misleading.

- Abstract: "Our results suggest that artificial intelligence models may be vastly more complicated than the proteins that they are modeling and that protein genetics is actually both simple and intelligible." and again p. 3: "It could be that the extreme complexity of these models reflects an underlying extreme complexity of protein genotype-phenotype landscapes. Alternatively, the actual genetic architecture of proteins might be much simpler than is evidenced by these models." - This is a confusing comparison and strange claim. The number of parameters in deep neural networks doesn't arise organically from the nature of genotype-phenotype landscapes; it reflects human choices for human applications. In many cases the number of parameters reflects the need for the network to generalize across many different proteins rather than encode one local fitness landscape for one wild-type sequence. I think this overall analogy between neural network complexity and fitness landscape complexity is unsound, like comparing meters to kilograms. Overall, the authors should reconsider what specific claims they want to make about neural networks, if any. No one would argue with the claim that "large neural network contains more parameters than the minimum required to model one protein's local fitness landscape to a particular degree of accuracy". But this defensible claim is also not controversial.

P. 3: "these models alone provide little insight into the actual genetic architecture of proteins." - Again this "little insight" claim is irrelevant, highly contestable and not supported by data in this paper. Using a deep neural network to predict 1st and 2nd order fitness effects could be very insightful.

P. 4: "nevertheless maintain abundance scores that are indistinguishable from that of the WT protein, nominal $P > 0.5$ " - What does this P value refer to? I don't see it in the methods.

P. 6: "Fitting linear and biophysical models to the combinatorial data improves performance by 30% and 13% respectively (Fig. 2b, upper panels)" - This refers to the additional variance captured by these models, not the "improved performance" (if a model with $R^2 0.5$ improves performance by 50%, does that mean $R^2 0.75$ or 1.0 ? I would clarify the language here).

Fig. 2b: It is slightly confusing that the main text narrative starts at the bottom of Fig. 2b rather than the top.

Fig. 2d: Both axes should be labeled "inferred" like in 2c.

Fig. 2d: The labeling is somewhat confusing because both axes show folding ddGs: the y-axis is inferred from the combinatorial data and the x-axis is inferred from the single and pairwise data. Could "combinatorial" and "single/pairwise" (or similar) be shown on the axes instead of "Folding" (which applies to both axes) and "ddPCA", which is a special term for one technique and not generally used?

Fig. 5k: It would help to label the plots on top "Inferred folding ddG" and "Inferred binding ddG", and label the y-axis "combinatorial library" and the x-axis "shallow double mutant library" or something like that. Again, both axes show "folding ddG" or "binding ddG"; the important thing is the data source so that is what should be shown on the axis label.

P. 14: "Energetic couplings show the same pattern, with folding coupling energies tending to be larger in magnitude than binding energetic couplings (AUC = 0.7, $n = 210$, $P = 3.6e-7$, two-sided Mann-Whitney U test, Fig. 6b)." - Give the means and standard deviations.

P. 17: The opening sentence of the first paragraph of discussion writes "at least some proteins" but then the concluding sentence states writes "proteins" (i.e. all proteins)? The unqualified statement does not seem supported, especially because the "understandable" part seems to hinge on Fig. 3f, when 3f would suggest there is actually much more that we don't understand than what we do. However, the second paragraph in discussion better supports the claim that this is a general property of proteins.

P. 17: "Analyses of previously-published combinatorial protein mutagenesis datasets 37,38, mutagenesis of a protein interaction interface 38, combinatorial mutagenesis of a tRNA 39 and mutagenesis of an alternatively spliced exon 35 suggest that this simplicity of genotype-phenotype landscapes is widely observed and likely to be a general principle of macromolecules and their molecular reactions." - Should probably cite refs 6 and 36 as well?

P. 17: "Energy models are grounded in our understanding of protein physics and their simplicity and interpretability contrasts with the extreme complexity and lack of mechanistic insight provided by deep neural networks." - This is confusing to me: is it fair to call the energetic terms inferred using the authors' approach "physical"? While a consistent physical model based on attractive and repulsive forces and entropic terms could be expected to be transferrable from one protein to another, the energetic terms inferred using the authors' approach are phenomenological and specific to one protein's fitness landscape. I like the authors' approach and think it is very elegant, but question whether we can say the approach is "grounded in our understanding of protein physics".

Referee #3:

Thank you for the opportunity to review “The genetic architecture of protein stability” by Faure et al. This manuscript discusses how biophysical models can explain the effect of genetic variation on protein structure and activity without a need to apply complex AI approaches using deep neural networks. Authors use combinatorial mutagenesis of GRB2-SH3 and measure abundance and binding using PCA screening approaches as well as calculate free energy of folding. They show that biophysical energy models that quantify free energy changes and energy couplings can accurately predict effects of mutations on proteins.

This study investigates how to best model mutational effects on protein structure and function. This addresses an important question in a number of fields investigating the genotypic and phenotypic variation, combinatorial genetics, genetic interactions and protein structure especially in face of emerging perceived importance of usefulness of the application of deep neural networks for biology. It provides novel insight about developing accurate models of combinatorial mutations on proteins. This is an elegant study and a tour de force.

Major comments

The authors provide compelling evidence for the use of linear models for accurate predictions of mutational effects on protein structure and function. They allude to the overfitting that would be introduced by AI models. However, they never actually apply any AI model in their study. It would be useful to compare the performance of an AI model with at least some of the results obtained here. Correlations with modeling predictions that are presented here are not 1 leaving a substantial portion of variation unexplained. An AI model can be called extremely difficult but that does not necessarily mean that it is not explainable. Predictions can be explained after model training to see how predictions of mutational effects compare to experimentally validated effects, which features are important and evaluate the statistical confidence in predictions.

Authors also do not provide any information about the coverage that they achieve with their mutagenesis experiments. What is the coverage of the mutagenesis? Is there a bias for certain residues to be mutated to certain others? Are all the combinations equally represented? These should be added as supplementary figures.

The limitation of this study is that it is a detailed assessment of one protein as a proof-of-concept. Other proteins that do not contain SH3 domains or those that are disordered may behave differently. This should be noted in the Discussion.

Minor comments

Fig. 1c is not cited in the main text.

Figures should be called out in order. For example, Fig. 4d is referenced before Fig. 4c.

Author Rebuttals to Initial Comments:

Referee #1:

Understanding how genotypes give rise to molecular and organismal phenotypes is arguably the central question of biology. Recent applications of machine learning to the problem have made major advances but may have limited interpretability. New sequencing-based high throughput genotype-phenotype assays have provided a wealth of datasets for probing this question as well. To date, these datasets have consisted of single substitutions, however, with a small number of double or higher-order combinations.

In this manuscript, Faure et al. measure the effects of combinatorial sets of mutations in several higher-order mutational scans to investigate how far biophysical models can be used to understand genotype-phenotype links. To avoid the combinatorial explosion in size that would result, the authors employ a clever and highly innovative 'Greedy' approach to design their library that harnesses their previous single substitution libraries (Faure, 2022, Nature) with the aim of maximizing the sampling of folded and functional mutants in their library. This approach is of significance and impact as it could help with efficient sampling of higher order combinatorial space.

One major open question they tackle is the nature of epistasis in these maps, i.e. non-additive free-energy changes in combinations of mutations, and particularly their sparsity (or how important progressively higher-order epistatic contributions are). Another open question is the complexity of these maps, and whether simple biophysical models are sufficient, or if more complex ML approaches are required, with the associated loss of direct mechanistically interpretable parameters. The authors, thorough analysis of their datasets, provide important answers to these questions, although the choice of their model system as a small protein domain limits the generalizability of their results to larger proteins with multiple folds.

This is a well-written, rigorous, thorough piece of work and will be of great interest to biologists broadly, with particular importance for protein biology, computational modeling, and evolutionary biochemistry communities. We feel that the argument and stakes could be sharpened, and there are some overstatements and important additional analyses that should be done. We recommend another round of revision at Nature after additional analyses are done.

We thank the referees for their enthusiasm and very constructive suggestions.

Major Comments

1. The authors emphasize they screen a mutational space of $\sim 10^{10}$, whereas prior work has only reached $\sim 10^6$. This is somewhat misleading. The authors do sample from a large design space, however they are still experimentally measuring $\sim 10^5$ variants. This is, as they admit, a small fraction of the overall space. They argue that the vast majority of mutations will produce a misfolded protein, which is likely valid, and so the space of informative genotypes is actually much smaller than 10^{10} . To make this claim, however, they use the predicted additive folding energies from their prior single substitution experiment. This seems to beg the question, in that they are assuming that the additivity of single variants is sufficient to predict gross higher-order phenotypes in order to avoid comprehensively measuring higher-order phenotypes, which they would do to prove that additivity is sufficient to predict higher-order phenotypes. Also, one could argue that approaches like error-prone PCR similarly sparsely sample extremely large sequence spaces. We recommend some discussion clarifying how the sampling strategy differs from other approaches, and walking back the claim that they are comprehensively sampling a mutational space of 10^{10} .

We agree and have clarified that we are ‘sampling’ very large sequence spaces throughout the manuscript.

2. The authors state throughout the results and in the conclusions that the data supports the notion that genetic landscapes and interactions are generally simple. We believe this claim requires more work. Although several obvious definitions for simplicity might apply, the authors do not explicitly define what they mean, so it is difficult to quantify the results here. Instead, they argue for simplicity by showing improved performance on prediction tasks using biophysical models with first or second-order interaction terms only. The saturation combinatorial mutational scan and the other higher order library both indeed show strong correlations with first- and second-order models (R^2 0.93 and 0.97). The Greedy dataset, however, has much lower correlation, giving an R^2 of 0.72 for the second-order model. Given that these fits apply to the same measurements on the same protein, the differences in correlation must have to do with the architecture of the different sequence spaces. The Greedy design, then, seems to create a space with increased importance of higher-order interactions, and so decreased simplicity in the terms of the authors. We recommend the authors improve their discussion of this implication, as well as more clearly state how “simplicity” is being determined. This may also help demonstrate the novelty of their technique, as the fact that most residues are not coupled has been described before going back to foundational work in combinatorial mutagenesis (Wells, 1990, Biochemistry, for example).

Thank you for raising this interesting point that was implied in the discussion of our original manuscript but not explicitly stated. We agree that the difference in predictive performance between the 15 dimensional sequence spaces ($R^2 = 0.97$ for Grb2-SH3, $R^2 = 0.87$ for SRC - new dataset described below) and the 34 dimensional sequence space ($R^2 = 0.72$) is suggestive of an

increasing importance of higher order genetic interactions (epistasis) as more mutations are combined. We have added a sentence to the discussion highlighting this: “Indeed the superior performance of our models in 2^{15} compared to in 2^{34} sized sequence spaces hints that higher order interactions become increasingly important as sequences diverge.”

We have also added a citation to the excellent and very useful Wells review and have added the following text to the introduction “For us, a simple model is one with few parameters (so providing a large data compression) and parameters that are interpretable (so providing understanding).”

3. The greedy mutational scanning approach is highly novel and is the first of its kind that explores much higher-order mutational space beyond 2nd-order. However, the authors do not explore combinatorial trajectories in depth beyond 2nd order effects - there should be further analysis and likely additional figure panels that explore effects beyond those that are 2nd order. Otherwise it is unclear why such a greedy deep mutational scan was done and why these analyses couldn't have been done on their existing data from Faure 2022 or why a more comprehensive double mutational scan couldn't have been done as has been previously done on GB2 (Olson, CA, 2014). In particular there may be interesting mutational trajectories that could be explored with this data if one contrasts the results with natural predicted evolutionary trajectories within SH3 orthologs or paralogs using similar analysis as has been done for enzymes with lower throughput combinatorial mutational experiments (Weinreich, 2006, Science). While this specific analysis is not necessary - it would be good to see further analyses of the higher order space.

We agree but believe further analysis of higher order interactions goes beyond the scope of the current manuscript. The combinatorial explosion of interactions to statistically test makes this a very non-trivial computational task that we believe requires further method development and probably different experimental designs.

4. In Fig. 3 the authors train models on structural features to predict mutational effects on a combinatorial dataset inferred using a 1st and 2nd order biophysical model built on the Greedy dataset. That the authors get a model based solely on protein features that performs about as well as Rosetta/Foldx approaches suggests something may be learned here about what determines epistasis. The authors find that backbone distance is important for predicting the effects of mutations whereas inter-residue distance is not. In other words, the placement in the primary sequence seems to be important while the 3D contacts do not. The authors refer to this observation a number of times throughout the manuscript including the following statement:

‘The observation that mutation effects can be modulated by the sequence context at distal sites and that the strength of this coupling depends in part on the number of intervening residues – but not 3D distance – suggests that energetic perturbations are particularly efficiently transmitted over long distances along the covalent bond structure of the peptide backbone.’

This is a very surprising result, and seems to contradict the typical understanding of what determines the folding of proteins. We therefore believe it requires further quantification and discussion. In SH3 domains, circular permutation, where a protein's N- and C-termini are swapped (which would change position in primary sequence but potentially maintain tertiary sequence), has been successfully performed (Viguera, 1996, Nat. Struct. Biol.). This appears to conflict with the broadest claims of the authors.

We also note that the features the authors include in the model may introduce structural information: information about pairwise residue interactions is encoded in their physicochemical interactions, which do provide informative features here. It is also possible that the inter-residue distance metric is not appropriate to describing the effects of a mutation, as the minimal heavy atom distance will change in mutant structures especially in the context of combinatorial mutations, which are the backgrounds that epistatic couplings would be occurring in.

Overall, that 3D contacts do not determine stability and backbone propagations is somewhat unorthodox and counterintuitive - this claim should be further theorized. An alternative helpful comparison that would be useful to contextualize the performance of the model in this manuscript would be to use structure-based effect predictors Rosetta/Foldx/Foldetta which will account for the structural perturbations of mutations and which could give a more reliable inter-residue distance as well (Gerasimavicius, Protein Science, 2023). Perhaps such a comparison would also be useful in guiding what features to include that are missing from the modeling currently - or in other words what the missing '0.5' predicted folding $\Delta\Delta G$ is. Currently the model's performance does not support the statement within the conclusion that the 'genetic architecture of proteins is predictable and understandable'.

We apologise for the confusion but our results are actually highly supportive of the hypothesis that structural contacts are a very important cause of energetic couplings (Fig. 3a, Fig. 7f). We have edited the sentence quoted above to avoid any confusion (in this library, all of the residues are structural contacts so the hypothesis is not tested). We agree that the decay of couplings along the protein backbone is interesting (and we also observe it in our new Src mutagenesis dataset, please see below). However, we also think that this is quite intuitive: local changes in structure and/or dynamics will, of course, be propagated to some extent via backbone movements.

5. Epistasis and interactions between residues within proteins are largely determined by the physicochemistry of residues. However, there is very little discussion currently that considers how physicochemistry of residues changes behavior of interacting residues and presence of non-additivity. For example, aromatic and non-polar burial is classically thought to drive folding - is there a core aromatic in SH3? If so, what is the effect of removing the aromatic and replacing it elsewhere

in the core? Similarly, ion interactions and charge swaps are the classic example for non-additivity - is this seen within the dataset? How does changing the distance between ionic pairs or locations in the protein affect folding/function? The richness of this dataset calls out for a specific physicochemical analysis.

We agree that better understanding the physicochemical determinants of energetic couplings is an interesting problem. However, binary combinatorial mutagenesis datasets are unfortunately not well designed to address this question as for each residue pair we have only quantified one of the 361 (=19x19) possible energetic couplings.

Minor Comments

1. Please add page and line numbers to help reviewers make specific references.

Done (+ apologies!).

2. The authors provide an excellent repository for the analysis and we would like to point that out. We would recommend they use Zenodo or a similar service to provide a reference to a specific release version, however.

Thank you for the suggestion. We have now used Zenodo to archive the first release (v1.0.0) of the GitHub repository (<https://github.com/lehner-lab/archstabms>) and issue a DOI for the archive: <https://zenodo.org/doi/10.5281/zenodo.11671164>

3. Page 3, paragraph 2: The authors cite Poelwijk (2019) and Sarkisyan (2016) for the claim that “combining random mutations in even moderate numbers nearly always results in unfolded proteins,” but these references don’t support that specific claim. Both references measure fluorescence phenotypes, rather than soluble or folded fractions, and so this claim should be adjusted.

Thank you. Edited to ‘non-functional’.

4. Page 3, paragraph 2: The authors should clarify that by “expected to be folded” they mean when assuming additivity, based on their previous single-site scans.

Thank you. Added.

5. Page 4, paragraph 4: The authors mention a “bottlenecked library,” but there are no other references to this.

Edited to ‘sampled’.

6. Page 10, paragraph 3: The authors mention testing the model on an ‘independent combinatorial’ experiment. What exactly is this dataset? Is this from the same research group and assay? Is it also a small protein?

Library 3, as now stated in the text.

7. Fig. 2f (and discussion thereof): These densities approximate the distributions of fitness effects of the subsampled combinatorial space, which was designed with the assumption of additivity. Do these densities fairly represent the entire space? Is it surprising that the epistatic couplings have mean 0?

That most energetic couplings are approximately zero is, we think, a key result of this study and is consistent with previous studies using smaller numbers of mutations (see e.g. Wells 1990): most non-contacting residues are not strongly energetically coupled. This is also observed in library 2 where there is no selection of mutations based on additivity but rather all possible mutations are combined in four positions, suggesting it is a general result.

8. Fig. 3a,b: What is the relationship between primary and tertiary sequence in SH3? Interpreting the difference in correlations between inter-residue distance and $\Delta\Delta\Delta G$ vs backbone distance and $\Delta\Delta\Delta G$ would be helped by understanding how inter-residue distance and backbone distance are themselves correlated.

This is presented in Fig. 3B where the inter-residue distance for each pair of mutations is indicated by colour. Moreover, library 2 presented in figure 4 is explicitly designed to address the effect of backbone distance when inter-residue distance is controlled: all pairs of residues but one in library 2 are contacts and there is still a quantifiable dependence on backbone separation.

9. Fig. 4i: The closest class of residues (i.e., 2 residues apart) appears to have significantly larger effects on folding $\Delta\Delta\Delta G$ than all others. We wonder how they are driving the correlation - what would the spearman rho be if they were excluded?

The closest class of residues (2 residues apart) indeed has larger coupling effects, but we still observe a significant anti-correlation with backbone distance when excluding this class (Spearman's $\rho = -0.25$, $P = 2.2e-16$).

10. It could be interesting to discuss whether different types of higher order libraries contain more or less folded proteins specifically. For instance, is the Greedy approach more or less enriched for folded proteins? Are there types of bootstrapping that could be used to more efficiently sample higher order space?

We agree this is a great question, but the four libraries differ in the magnitude of the individual mutational effects and only one of the libraries is a non-greedy library, so it would be dangerous to make any conclusions beyond the obvious conclusion that greedy libraries are much more enriched for folded proteins than random sampling (e.g. ED Fig 1a).

11. Adding a short definition of “genetic architecture” would be useful for non-specialists.

Now defined in paragraph 2 of the introduction.

12. We would also suggest a bit more care with the language of “energetic perturbations” being “transmitted over long distances along the covalent bond structure of the peptide backbone,” and similar – this, to me, suggests some physical mechanism or interaction (a causation) that is not necessarily being claimed (there is a clear correlation, however).

We agree - thank you.

13. Additional care to the use of “generalizability” and similar language should be taken. A careless reader might interpret this to mean generalizability across proteins, whereas here it means generalizability across orders of mutations.

We agree - thank you.

14. There are a few important citations that could help contextualize this for the reader. On the additivity of mutational effects, Wells (1990, Biochemistry) is an important foundational reference. The work of Michael Harms, and particularly his group’s approach to linearizing genotype-phenotype maps, should be addressed. Dryden, et al. (2008, J. R. Soc. Interface) is an important work for putting sequence space sizes into perspective as well.

We have added these citations - many thanks.

Referee #2:

This paper explores the determinants of folding and binding for an SH3 domain of the protein GRB2. The authors experimentally measure folding and binding for several large libraries of combinatorial mutants. They then use their previously developed approach to infer the individual effects of single mutants as well as pairwise couplings between residues. The authors find that relatively sparse models (single and pairwise terms only) can model the overall fitness landscape very well, even in sequences far away from wild-type (>15 mutations away). This is noteworthy, but somewhat unsurprising: as the authors note, similar studies have come to similar conclusions (paragraph 2 of discussion). However, the number of mutations explored is much larger in this study than in previous work.

The main strengths of the paper are:

1. The paper explores a very large combinatorial sequence space (simultaneous mutations at 34 positions for folding and 14 positions for binding). This scale is exciting and novel, especially in the context of folding stability (most large-scale studies examine a functional phenotype such as binding or fluorescence).
2. The paper simultaneously infers coupling effects for both folding and binding on a more limited library scale (mutations at 14 sites). As the authors note, I think this is the first example of inferring couplings for both phenotypes on a large scale. These results add to our understanding of sequence-phenotype relationships and also provide valuable data for testing machine learning models.
3. The finding that folding couplings more strongly correlate with primary sequence separation rather than structural separation is surprising and noteworthy.
4. The paper is very clearly written with figures that are mostly easy to understand (with some exceptions below).

Overall the conclusions of the study are well supported by their data (with minor exceptions noted below). I think the study will be exciting and interesting to protein biophysicists and readers interested in the high-throughput technology and machine learning in protein science.

We thank the referee for their enthusiasm and very constructive suggestions.

The main weaknesses of the paper are:

1. The abstract, introduction, and discussion contain many irrelevant (and arguably incorrect) claims about deep neural networks. It does not make sense to me to compare the number of parameters in a neural network to the number of epistatic terms required to model one protein's fitness landscape. Analogously, the human brain is a very complex neural network, but can still solve very simple math problems - this doesn't mean the brain is "too complicated". I would suggest removing all the discussion about neural networks, or at minimum revising the claims to be more specific and defensible (see below).

We agree.

2. The abstract states that the authors "experimentally explore sequence spaces $>10^{10}$ ". This is misleading because it suggests that 10^{10} sequences were actually experimentally examined, when the actual number is 10^5 . I think the exploration of mutations at 34 sites is noteworthy and important, and perhaps this is a more accurate way of describing what was performed. Or perhaps "By experimentally examining 100,000 sequences out of a possible space of 10^{10} ..."

We agree and have made it clear that we are experimentally exploring random samples out of these spaces.

3. It is somewhat unclear how "physical" the authors model is. They refer to it as "physical" many times, and the parameters of their model are interpretable. However, we would expect truly physical models to be transferrable from one protein to another. In contrast, the author's model is phenomenological and only valid for this one protein. Perhaps different terminology (or a clearer definition of what makes the model physical) would be beneficial.

We agree and refer to the models as 'energy models' and 'thermodynamic models' in the revised manuscript.

Other clarifications and suggestions to the author:

- Abstract: "However, these models are extremely complicated and provide little insight into the fundamental genetic architecture of proteins." - The amount of insight that can be gained from deep neural networks is very contentious and the "little insight" claim in the abstract is very debatable. A less contestable claim would be that individual parameters in neural networks don't have physical interpretations (but model predictions do have physical interpretations).

We have edited this sentence to read: "However, these models are extremely complicated and challenging to understand, to-date providing little insight into the fundamental genetic architecture of proteins."

- Abstract: "Here, by experimentally exploring sequence spaces $>10^{10}$ " - As above, this is misleading.

Edited to read "Here, by experimentally sampling from..".

- Abstract: "Our results suggest that artificial intelligence models may be vastly more complicated than the proteins that they are modeling and that protein genetics is actually both simple and intelligible." and again p. 3: "It could be that the extreme complexity of these models reflects an underlying extreme complexity of protein genotype-phenotype landscapes. Alternatively, the actual genetic architecture of proteins might be much simpler than is evidenced by these models." - This is a confusing comparison and strange claim. The number of parameters in deep neural networks doesn't arise organically from the nature of genotype-phenotype landscapes; it reflects human choices for human applications. In many cases the number of parameters reflects the need for the network to generalize across many different proteins rather than encode one local fitness landscape for one wild-type sequence. I think this overall analogy between neural network complexity and fitness landscape complexity is unsound, like comparing meters to kilograms. Overall, the authors should reconsider what specific claims they want to make about neural networks, if any. No one would argue with the claim that "large neural network contains more parameters than the minimum required to model one protein's local fitness landscape to a particular degree of accuracy". But this defensible claim is also not controversial.

We mostly agree and have deleted the quoted phrase from the abstract and re-worded the sentences from the introduction to avoid the direct comparison. (However, as an aside, we do find it alarmingly widespread that datasets containing nothing measurable beyond simple sigmoidal global epistasis are used to demonstrate the utility of various ML approaches for protein engineering i.e. the lack of sensible baseline models in many studies.)

P. 3: "these models alone provide little insight into the actual genetic architecture of proteins." - Again this "little insight" claim is irrelevant, highly contestable and not supported by data in this paper. Using a deep neural network to predict 1st and 2nd order fitness effects could be very insightful.

Edited to read: "However, these models have extremely complicated and difficult to interpret architectures."

P. 4: "nevertheless maintain abundance scores that are indistinguishable from that of the WT protein, nominal $P > 0.5$ " - What does this P value refer to? I don't see it in the methods.

The P value refers to the result of a two-sided z-test comparing the observed abundance scores (fitness estimates and associated errors of each variant) to that of the WT protein. We have now indicated the corresponding test alongside the nominal P value threshold.

P. 6: "Fitting linear and biophysical models to the combinatorial data improves performance by 30% and 13% respectively (Fig. 2b, upper panels)" - This refers to the additional variance captured by these models, not the "improved performance" (if a model with R^2 0.5 improves performance by 50%, does that mean R^2 0.75 or 1.0? I would clarify the language here).

Thank you - edited.

Fig. 2b: It is slightly confusing that the main text narrative starts at the bottom of Fig. 2b rather than the top.

We have not changed the panel arrangement, but can do so if required.

Fig. 2d: Both axes should be labeled "inferred" like in 2c.

Fig. 2d: The labeling is somewhat confusing because both axes show folding $\Delta\Delta G$ s: the y-axis is inferred from the combinatorial data and the x-axis is inferred from the single and pairwise data. Could "combinatorial" and "single/pairwise" (or similar) be shown on the axes instead of "Folding" (which applies to both axes) and "ddPCA", which is a special term for one technique and not generally used?

Axes labels changed to 'Inferred ddG (this study, kcal/mol) and 'Inferred ddG (double mutants, kcal/mol)'.

Fig. 5k: It would help to label the plots on top "Inferred folding ddG" and "Inferred binding ddG", and label the y-axis "combinatorial library" and the x-axis "shallow double mutant library" or something like that. Again, both axes show "folding ddG" or "binding ddG"; the important thing is the data source so that is what should be shown on the axis label.

We have made these changes to Fig. 5k to clarify the data source.

P. 14: "Energetic couplings show the same pattern, with folding coupling energies tending to be larger in magnitude than binding energetic couplings (AUC = 0.7, n = 210, P = 3.6e-7, two-sided Mann-Whitney U test, Fig. 6b)." - Give the means and standard deviations.

We have now added the means and standard deviations.

P. 17: The opening sentence of the first paragraph of discussion writes "at least some proteins" but then the concluding sentence states writes "proteins" (i.e. all proteins)? The unqualified statement does not seem supported, especially because the "understandable" part seems to hinge on Fig. 3f, when 3f would suggest there is actually much more that we don't understand than what we do. However, the second paragraph in discussion better supports the claim that this is a general property of proteins.

We have deleted the last sentence of paragraph one of the discussion.

P. 17: "Analyses of previously-published combinatorial protein mutagenesis datasets 37,38, mutagenesis of a protein interaction interface 38, combinatorial mutagenesis of a tRNA 39 and mutagenesis of an alternatively spliced exon 35 suggest that this simplicity of genotype-phenotype landscapes is widely observed and likely to be a general principle of macromolecules and their molecular reactions." - Should probably cite refs 6 and 36 as well?

Agree - we have added the references.

P. 17: "Energy models are grounded in our understanding of protein physics and their simplicity and interpretability contrasts with the extreme complexity and lack of mechanistic insight provided by deep neural networks." - This is confusing to me: is it fair to call the energetic terms inferred using the authors' approach "physical"? While a consistent physical model based on attractive and repulsive forces and entropic terms could be expected to be transferrable from one protein to another, the energetic terms inferred using the authors' approach are phenomenological and specific to one protein's fitness landscape. I like the authors' approach and think it is very elegant, but question whether we can say the approach is "grounded in our understanding of protein physics".

We have edited this sentence to read 'thermodynamics' rather than 'physics'.

Referee #3:

Thank you for the opportunity to review “The genetic architecture of protein stability” by Faure et al. This manuscript discusses how biophysical models can explain the effect of genetic variation on protein structure and activity without a need to apply complex AI approaches using deep neural networks. Authors use combinatorial mutagenesis of GRB2-SH3 and measure abundance and binding using PCA screening approaches as well as calculate free energy of folding. They show that biophysical energy models that quantify free energy changes and energy couplings can accurately predict effects of mutations on proteins.

This study investigates how to best model mutational effects on protein structure and function. This addresses an important question in a number of fields investigating the genotypic and phenotypic variation, combinatorial genetics, genetic interactions and protein structure especially in face of emerging perceived importance of usefulness of the application of deep neural networks for biology. It provides novel insight about developing accurate models of combinatorial mutations on proteins. This is an elegant study and a tour de force.

We thank the referee for their enthusiasm and very constructive suggestions.

Major comments

The authors provide compelling evidence for the use of linear models for accurate predictions of mutational effects on protein structure and function. They allude to the overfitting that would be introduced by AI models. However, they never actually apply any AI model in their study. It would be useful to compare the performance of an AI model with at least some of the results obtained here. Correlations with modeling predictions that are presented here are not 1 leaving a substantial portion of variation unexplained. An AI model can be called extremely difficult but that does not necessarily mean that it is not explainable. Predictions can be explained after model training to see how predictions of mutational effects compare to experimentally validated effects, which features are important and evaluate the statistical confidence in predictions.

We agree and, as detailed above in response to referee 2, we have removed these textual contrasts with AI models.

Authors also do not provide any information about the coverage that they achieve with their mutagenesis experiments. What is the coverage of the mutagenesis? Is there a bias for certain residues to be mutated to certain others? Are all the combinations equally represented? These should be added as supplementary figures.

For library 1, we obtained triplicate abundance measurements for 129,320 variants, which is 0.0007% of the sequence space, and the number of genotypes in the sample matching the expected pod-like shape of the genotype frequency landscape, with the number of genotypes peaking at the intermediate Hamming distance of 17 i.e. equidistant from the WT (0th order) and 34th order mutant. For library 2 we made abundance measurements for 138,157 variants, which is 86% of the sequence landscape. For library 3 we obtained binding measurements for 25,967 variants and

abundance measurements for 31,936 variants (79% and 97% of the sequence landscape, respectively). These numbers are all given in the results section of the text.

The limitation of this study is that it is a detailed assessment of one protein as a proof-of-concept. Other proteins that do not contain SH3 domains or those that are disordered may behave differently. This should be noted in the Discussion.

In the revised manuscript we have included data from an additional combinatorial mutagenesis selection experiment where we used the same greedy approach to design a library containing 2^{15} (=32,768) variants in an unrelated and much larger protein, the human SRC oncoprotein. We obtained triplicate abundance measurements for 31,557 variants and the measurements were highly reproducible ($r > 0.86$). Fitting a second order energy model to this dataset gave results that were strikingly consistent with those that we observed with our three GRB2-SH3 libraries: the energy model is highly predictive ($R^2 = 0.87$), with sparse energetic couplings predicted by both close 3D spatial proximity and couplings decaying along the linear sequence. This data is presented in Fig. 7 and further supports the generality of our conclusions. As noted in the discussion, combinatorial mutagenesis of additional globular proteins, hydrophobic protein cores, an intrinsically disordered region, and RNAs have also revealed the high predictive performance of additive energy models with sparse pairwise energetic couplings.

Minor comments

Fig. 1c is not cited in the main text.

Thank you - now cited.

Figures should be called out in order. For example, Fig. 4d is referenced before Fig. 4c.

Our understanding is that figures but not necessarily figure panels need to be cited in order in the text. We can of course change the panel ordering, if required.

Reviewer Reports on the First Revision:

Referee #1:

We appreciate the changes and clarifications made by the authors in response to our initial review, and we are satisfied with the manuscript in its current form and believe it has been significantly improved. We have one remaining minor comment below, but do not believe this necessarily calls for revision.

Minor comment:

We found the response to our initial minor point 7 (re: Fig. 2f) particularly interesting. We agree that the distributions in library 2 are consistent with those of library 1; however, the variance is larger, and the dddG distribution appears to have a skew. This makes us wonder to what extent this reflects the greedy design process rather than the total distribution of the sampled space. Inclusion of the plot of library 2's density from this rebuttal in the extended data would be helpful for the reader to understand, we believe.

Referee #2:

I am satisfied with the changes made by the authors, with some caveats that I suggest should be discussed between the author and the editor.

1. The abstract states "However, these models are extremely complicated and challenging to understand, to-date providing little insight into the fundamental genetic architecture of proteins". This is new language based on my earlier comment to change/remove the discussion of neural networks. The question of whether deep neural networks have provided important insights into the genetic architecture is a matter of opinion, and I don't agree with the authors' opinion on this point. I would not claim this and don't think it's necessary for the paper, but it's their paper and not mine.

2. I don't think it's appropriate to say in the abstract that "These energetic couplings are sparse and predicted by structural contacts and backbone proximity." Energetic couplings occur at proximal sites, but nothing in the paper predicts the actual value of these couplings. Predicting that there may be a coupling is different from predicting the value of the coupling itself, and the correlations in Fig. 3a,b are very low. I think "These energetic couplings are sparse and associated with structural contacts and backbone proximity" (or similar) would be more appropriate. "Predicted by" seems to be new language replacing the original "caused by", but I think "predicted by" implies something the authors don't mean to imply.

Referee #3:

The authors addressed sufficiently most comments. However, all figures and figure panels should be called out sequentially in the main text. For example, Fig. 4d is still referenced before Fig. 4c.

Author Rebuttals to First Revision:

Referee #1:

We appreciate the changes and clarifications made by the authors in response to our initial review, and we are satisfied with the manuscript in its current form and believe it has been significantly improved. We have one remaining minor comment below, but do not believe this necessarily calls for revision.

Minor comment:

We found the response to our initial minor point 7 (re: Fig. 2f) particularly interesting. We agree that the distributions in library 2 are consistent with those of library 1; however, the variance is larger, and the dddG distribution appears to have a skew. This makes us wonder to what extent this reflects the greedy design process rather than the total distribution of the sampled space. Inclusion of the plot of library 2's density from this rebuttal in the extended data would be helpful for the reader to understand, we believe.

We have now added the plot of library 2's folding free energy changes and pairwise energetic couplings to Extended Data Fig. 5.

Referee #2:

I am satisfied with the changes made by the authors, with some caveats that I suggest should be discussed between the author and the editor.

1. The abstract states "However, these models are extremely complicated and challenging to understand, to-date providing little insight into the fundamental genetic architecture of proteins". This is new language based on my earlier comment to change/remove the discussion of neural networks. The question of whether deep neural networks have provided important insights into the genetic architecture is a matter of opinion, and I don't agree with the authors' opinion on this point. I would not claim this and don't think it's necessary for the paper, but it's their paper and not mine.

We have changed this sentence in the abstract to: "However, these models are extremely complicated."

2. I don't think it's appropriate to say in the abstract that "These energetic couplings are sparse and predicted by structural contacts and backbone proximity." Energetic couplings occur at proximal sites, but nothing in the paper predicts the actual value of these couplings. Predicting that there may be a coupling is different from predicting the value of the coupling itself, and the correlations in Fig. 3a,b are very low. I think "These energetic couplings are sparse and associated with structural contacts and backbone proximity" (or similar) would be more appropriate. "Predicted by" seems to be new language replacing the original "caused by", but I think "predicted by" implies something the authors don't mean to imply.

We have changed the sentence according to the reviewer's recommendation: "These energetic couplings are sparse and associated with structural contacts and backbone proximity."

Referee #3:

The authors addressed sufficiently most comments. However, all figures and figure panels should be called out sequentially in the main text. For example, Fig. 4d is still referenced before Fig. 4c.

We have fixed the figure/panel order throughout.